

# Late Holocene evolution of a coupled, mud-dominated delta plain–chenier plain system, coastal Louisiana, USA

Marc P. Hijma[1,2], Zhixiong Shen[1,3], Torbjörn E. Törnqvist[1], Barbara Mauz[4]

[1]Department of Earth and Environmental Sciences, Tulane University, 6823 St. Charles Avenue, New Orleans, Louisiana
70118-5698, USA
[2]Department of Applied Geology and Geophysics, Deltares, P.O. Box 85467, 3508 AL Utrecht, The Netherlands
[3]Department of Marine Science, Coastal Carolina University, PO Box 261954, Conway, South Carolina 29528, USA
[4]Department of Geography and Planning, University of Liverpool, Liverpool L69 7ZT, UK

*Correspondence to*: Marc P. Hijma (marc.hijma@deltares.nl)

**Abstract.** Major deltas and their adjacent coastal plains are commonly linked by means of coast-parallel fluxes of water, sediment, and nutrients. Observations of the evolution of these interlinked systems over centennial to millennial timescales are essential to understand the interaction between point sources of sediment discharge (i.e., deltaic distributaries) and adjacent coastal plains across large spatial (i.e., hundreds of kilometres) scales. This information is needed to constrain

future generations of numerical models to predict coastal evolution in relation to climate change and other human activities. Here we examine the coastal plain adjacent to the Mississippi River Delta, one of the world's largest deltas. We use a refined chronology based on 22 new optically stimulated luminescence and 22 new radiocarbon ages to test the hypothesis that cyclic Mississippi subdelta shifting has influenced the evolution of the adjacent Chenier Plain (CP). We show that over the past 3 kyr, accumulation rates in the CP were generally 0-1 MT yr$^{-1}$. However, between 1.2 and 0.6 ka, when the Mississippi

River shifted to a position more proximal to the CP, these rates increased to 2.9 ± 1.1 MT yr$^{-1}$ or 0.5-1.5% of the total sediment load of the Mississippi River. We conclude that CP evolution during the past 3 kyr was partly a direct consequence of shifting subdeltas, in addition to changing regional sediment sources and modest rates of relative sea-level rise. These findings have implications for Mississippi River sediment diversions that are currently being planned to restore portions of this vulnerable coast. Only if such diversions are planned in the western portion of the Mississippi Delta Plain they could

potentially contribute to sustaining the CP shoreline.



# 1 Introduction

Low-elevation coastal zones are facing severe pressures due to a combination of rapid coastal development (e.g. McGranahan et al., 2007), the effects of accelerated relative sea-level (RSL) rise (e.g. Ericson et al., 2006), and sediment deficits (e.g. Syvitski et al., 2005). The steadily increasing proportion of the world population in coastal lowlands has

5  become one of the most pressing global environmental problems within the context of climate change (Wong et al., 2014). This is particularly the case for major deltas and their adjacent coastal plains that are linked by means of coast-parallel fluxes of water, sediment, and nutrients. Mud constitutes a dominant component of this material flux as exemplified by some of the world's largest sediment-delivery systems (e.g. Saito et al., 2000; Anthony et al., 2013; Szczuciński et al., 2013), yet surprisingly few studies have focused on the large-scale evolution of mud-dominated shorelines.

Observations over centennial to millennial timescales are particularly useful to understand the interaction between point sources of sediment discharge (i.e., deltaic distributaries) and adjacent coastal plains across large spatial (i.e., hundreds of kilometres) scales. The Holocene stratigraphic record contains a potentially powerful but underutilized archive for this purpose. In addition to increasing our understanding of large-scale coastal morphodynamics, information from the Holocene

record is essential to constrain future generations of numerical models that will be needed to enable predictions about coastal evolution (e.g. Allison and Meselhe, 2010; Paola et al., 2011). Such models can be expected to gain in importance in view of the enormous investments in coastal management and restoration efforts that will likely be implemented around the globe.

The Mississippi River has constructed one of the world's largest delta plains (the Mississippi Delta Plain, MDP) during the

Holocene. The MDP is presently dissected by two major distributaries (the Mississippi River and the Atchafalaya River) which feed active parts of the delta referred to herein as subdeltas (cf. Russell, 1940). In the past, the Mississippi River has shifted its course periodically as is evident from abandoned (inactive) subdeltas (Fig. 1). The associated redistribution of sediment along the coast resulted in 'healing' of scars in the coastline. Currently, the distributaries are completely embanked, resulting in large, sediment-starved sections that subside and erode rapidly. Coastal Louisiana experiences among the

world's highest rates of wetland loss, estimated at about 40 km$^2$ yr$^{-1}$ from 1985 to 2010 (Couvillion et al., 2011). This coastal degradation could be mitigated by artificially diverting sediment from the river back to the MDP (Day et al., 2007) which could potentially also influence the evolution of the Chenier Plain (CP), farther to the west (Fig 1). The CP is a 250 km long and 20-40 km wide low-lying marsh area with interspersed sandy ridges (cheniers). It has been proposed that during the past three millennia (Gould and McFarlan, 1959), several cycles of Mississippi subdelta shifting resulted in the formation of

alternating cheniers and mudflats (Russell and Howe, 1935; McBride et al., 2007). The hypothesis is that when the mouth of the Mississippi River is situated close to the CP, large amounts of muddy sediment are transported towards the CP via the east to west longshore current. When the river mouth shifts to a more easterly position, mud delivery is reduced and waves can attack and rework the mudflats, hereby forming the cheniers (Russell and Howe, 1935). To test this hypothesis it is





essential to have proper time control for the active period of past subdeltas as well as for the formation of the cheniers. At present this time control is still largely based on research and radiocarbon ages from the 1950s-1960s (Gould and McFarlan, 1959; McFarlan, 1961; Saucier, 1963; Frazier, 1967). The chronology of the cheniers is based on reworked shells that could predate the cheniers considerably (e.g. Shang et al., 2016). Over the past half century, the accuracy of radiocarbon dating and sampling strategies have increased significantly. For instance, re-examining one of the shifts of the Mississippi River (Törnqvist et al., 1996) resulted in an age that differed up to 2000 radiocarbon years from previously established ages. A major step forward in the last few decades is the possibility to directly determine the age of deposition of clastic sediments using optically stimulated luminescence (OSL). In recent years this method has been successfully applied to date sand and very fine silt in MDP sediments (Shen and Mauz, 2012; Shen et al., 2015; Shen et al., 2017).

Here we present new chronological data for both the CP and MDP to more rigorously test the hypothesis that their evolution was interlinked. For the sandy cheniers we used OSL to date their period of formation. In the MDP we used radiocarbon and OSL dating to refine the existing chronology. We traced six major chenier paleo-shorelines and calculated the area and mass of the interspersed mudflats to estimate minimum sediment accumulation rates through time. We aimed to determine to what extent the evolution of the CP is linked to subdelta shifting in the MDP, including the possible implications for coastal restoration plans. In addition, we examined the relationship between chenier formation and late Holocene RSL rise.

## 2 Regional setting and previous research

### 2.1 Chenier Plain

The northern border of the CP is formed by the outcropping Prairie Allogroup (Heinrich, 2006) that dips towards the south and is onlapped by Holocene strata (Fig. S1). The Pleistocene headlands reach farthest south at the location of the Lafayette meander belt (Fig. 1a) that was dated to Marine Isotope Stage 5a (Shen et al., 2012). The CP consists of widespread marshes with interspersed ridges that constitute the only dry, habitable areas. They are oriented roughly parallel to the current shoreline, have mean elevations of 1-2 m NAVD 88 (all elevations in this paper are with respect to the North American Vertical Datum (NAVD) of 1988, roughly equivalent to present day mean sea level) and can have lengths of tens of kilometers. The width of the ridges varies considerably due to overwash deposits and the presence of merging ridges, but is ~200 m on average. Most of the ridges are cheniers, meaning that they are "beach ridges, resting on silty or clayey deposits, which become isolated from the shore by a band of tidal mudflats" (Otvos and Price, 1979) and "flanked by intervening and usually wider intertidal-subtidal flats" (Otvos, 2000). Cheniers form when progradation is interrupted by a phase of erosion and transgression and mainly consist of (very) fine sand and shells due to winnowing processes. Once formed, they usually migrate landward due to washover process until the crest becomes high enough to withstand the highest spring tides (Augustinus, 1989). From that point on they are rather stable, accretionary features that sometimes start to prograde seaward (Gould and McFarlan, 1959) and become regressive cheniers (cf. Otvos, 2000). Our study focuses on the central part of the



CP (Fig. 1) as it contains the most complete series of cheniers. In addition to cheniers, some ridges in the CP, especially around river mouths, started as spits and built out laterally as curved beach ridges (Gould and McFarlan, 1959; Penland and Suter, 1989; McBride et al., 2007). The dominant onshore wave approach is from the southeast, resulting in a longshore current to the west (Fig. 1), although ridge morphology near river mouths show clear signs of local reversal due to ebb-tidal

estuarine interactions (McBride et al., 2007). Four small rivers dissect the CP (Rosen and Xu, 2011): Sabine River (average discharge of 219 $m^3$ $s^{-1}$), Calcasieu River (72 $m^3$ $s^{-1}$), Mermenteau River (82 $m^3$ $s^{-1}$) and Vermillion River (33 $m^3$ $s^{-1}$). The mean tidal range is on the order of 0.3-0.4 m and is unlikely to have seen much change over the time window of interest to the present study (Hill et al., 2011).

The first and still the most extensive set of cross sections across the CP was presented by Fisk (1948), with considerable detail added by Byrne et al. (1959). Together with Gould and McFarlan (1959) who used extensive radiocarbon dating (Table S1) to reconstruct the geological history, these three papers still form the nucleus for our understanding of CP evolution. Above the Pleistocene substrate, Gould and McFarlan (1959) recognized a transgressive sequence extending all the way to the Pleistocene outcrops north of the CP (Fig. S1). Penland and Suter (1989) noted that the absence of clear

shoreline features along the Pleistocene outcrop and the presence of thick marsh deposits between the Pleistocene outcrop and the most landward cheniers make it unlikely that the shoreline ever reached the outcrop itself. The cheniers contain numerous shells or shell fragments, sometimes concentrated in shell hash. The seaward front of the cheniers is steep (3-7%), while the landward side is gentle, grades into the marsh and was formed during overwash events. Combining maps and radiocarbon dating (Table S1), Gould and McFarlan (1959) identified several paleo-shorelines of which the Little Chenier,

Creole-Pumpkin Ridge, Oak Grove-Grand Chenier and the chenier near the present shoreline are the most prominent (Fig. 2). They concluded that the CP formed during the past ~3 kyr as a result of net progradation. Yu et al. (2012) showed that RSL in the CP was about 1.5 m below present mean sea level around 3 ka, thus challenging the hypothesis (Penland and Suter, 1989; McBride et al., 2007) that RSL fall was one of the drivers of CP progradation.

**2.2 Mississippi Delta Plain**

Mississippi subdelta shifting during the Holocene has been studied intensively during the last century, resulting in a robust stratigraphic framework (see Blum and Roberts, 2012 for a recent review). The five most recent subdeltas (Teche, St. Bernard, Lafourche, Plaquemines-Modern, Atchafalaya; Fig. 1b) formed during a period of continuous RSL rise (González and Törnqvist, 2009; Yu et al., 2012), are generally well preserved and hence mapping has been reasonably straightforward

(see e.g. Roberts and Coleman, 1996). The chronology of these subdeltas is still largely based on work from the 1960s (McFarlan, 1961; Saucier, 1963; Frazier, 1967), although later work has led to significant revisions (Penland et al., 1987; Autin et al., 1991; Törnqvist et al., 1996). The subdeltas generally formed in less than 10 m deep water, with the exception of the currently active Plaquemines-Modern subdelta that has prograded into relatively deep water (> 50 m); its mouth is




situated close to the shelf edge (Fisk et al., 1954). The combined sediment delivery to the Gulf of Mexico by the Mississippi and Atchafalaya Rivers is about 175 MT yr[-1] (Meade and Moody, 2010). This is considerably lower than the 400-500 MT yr[-1] right before upstream parts of the Mississippi River were dammed and the estimated average of 230-290 MT yr[-1] for the last 12 kyr (Blum and Roberts, 2009). For our calculations we assume that the total sediment load of the Mississippi River

during CP-evolution was somewhere between 200-400 MT yr[-1]. As in the CP, the mean tidal range along the MDP is 0.3-0.4 m.

## 2.3 Conceptual models of interlinked Chenier Plain and Mississippi Delta Plain evolution

The Mississippi River mud is transported westward by the longshore current and forms a blanket on the shelf. Mudflat

accretion on the CP is linked to high-energy events (cold front passages, storms) when the mud is transported onshore (Roberts et al., 1989; Draut et al., 2005a). It has long been assumed (Howe et al., 1935; Russell and Howe, 1935) that when the western part of the MDP (within 100 km from the CP) is active, more mud can reach the CP than when the eastern part is active (the present Mississippi River mouth lies ~350 km from the CP). Similar inferences have been made for other major delta regions that host cheniers (Saito et al., 2000; Anthony et al., 2013). Recent mudflat accretion immediately west of the

Atchafalaya River mouth exemplifies that parts of the sediment output of the MDP end up in the CP (Draut et al., 2005b). Gould and McFarlan (1959), however, already indicated that this relationship is not straightforward and described periods with simultaneous mudflat accumulation and chenier formation in different portions of the CP. Likewise, McBride et al. (2007) reported the simultaneous growth of transgressive, regressive and laterally-accreted ridges. They agreed in general with the model proposed in the 1930s, but highlighted that during the transgressive phase of chenier formation, regressive

ridges can form near stable river outlets and laterally-accreted ridges near unstable outlets.

The two most recent papers addressing the CP-MDP link (Penland and Suter, 1989; McBride et al., 2007) correlate CP erosion/progradation patterns to bifurcations within the Lafourche subdelta. In addition to changes in the MDP, McBride et al. (2007) suggest that the formation of the Little Chenier and the Grand Chenier paleo-shorelines is linked to periods of

higher than present-day sea levels. Other potential factors influencing chenier formation are climatic changes, storm frequency, wave- and tidal regime changes and bay geometry (Augustinus, 1989). This shows that when studying the sensitivity of the CP to changes in the MDP, the influence of these latter changes have to be separated from more local influences on CP formation. At present, progress on this problem is held back by the lack of robust chronologies.



# 3 Materials and methods

## 3.1 Stratigraphy and sampling

Five clearly defined and widely spaced cheniers just west of the Mermentau River were studied (Fig. 2): Oak Grove Ridge, Pumpkin Ridge, Mura Ridge, Chenier Perdue and Little Chenier. We cored several cross sections to understand the local

stratigraphy (Figs. 3 and 4) using an Edelman auger and a 1-m-long gouge with 3 cm diameter.All the sediments were described in the field according to the US Department of Agriculture texture classification system. We classified the depositional environment either as chenier or as non-chenier. The cheniers were labelled as such based on their geomorphological expression, their stratigraphic position above fine-grained sediments and their sedimentological characteristics (sand and shells). Our deepest boreholes reach the Pleistocene substrate that is very stiff and mottled and

hence easily recognizable. The most sandy and homogenous parts of the cheniers, mostly in the center, were chosen for OSL sampling. The 2σ-error range of the OSL-ages is in the order of 200-600 yr and since the active period of cheniers is relatively short, this range will cover the period of existence of the cheniers and hence we assume that the OSL-ages are representative for the period of formation of the cheniers.

For OSL sampling, we first drilled with the Edelman auger to right above the targeted level and then attached an Eijkelkamp liner sampler, a 30-cm-long and 5-cm-wide metal cylinder with a plastic liner, to the extension rods. This cylinder was then hammered into the ground. Once lifted and detached, the liner sampler was extruded within a light-tight, black plastic bag.. Surface elevations were obtained using DEM data (Gesch, 2007; LSU, 2011) with a vertical accuracy of about 0.25 m. DEM data were also used to plot the land surface in the cross sections. The geographical positions of borehole sites were

determined using a hand-held GPS (accuracy 5-10 m).

To improve the chronology of the MDP we focused on constraining periods of activity of the trunk channels that feed the Teche, St. Bernard and Plaquemines-Modern subdeltas, but also dated some smaller distributaries that occur within these subdeltas. Using the same equipment as in the CP, multiple cross sections were again constructed before sampling.

Depending on the proximity to the main channel, they exhibit a sandy channel belt with adjacent natural-levee deposits consisting of silt loam and silty clay loam. Moving further into the flood basin, silty clay and clay become dominant and humic clay layers occur frequently. In most cases a peat bed occurs below the overbank deposits, although below the proximal natural-levee deposits peat is often eroded. Sometimes the overbank deposits are covered by a paleosol that passes into a peat bed in the flood basin. The beginning of subdelta activity was dated by sampling the top of peat beds below the

overbank deposits of the trunk channel, whereas the end of activity was constrained by dating the base of peat beds overlying the overbank deposits. The radiocarbon samples were taken with a 6-cm-wide gouge. As significant amounts of time can elapse before peat starts to form after channel abandonment (Törnqvist and Van Dijk, 1993), we also dated the top of natural-levee deposits using OSL to better constrain the period of activity.



## 3.2 Dating

### 3.2.1 OSL dating

Quartz OSL dating is a dosimetric technique that measures typically the time when quartz was latest exposed to sunlight (Aitken, 1998) and has an upper age limit of about 200 ka (Rhodes, 2011). Therefore, it is very useful for dating clastic-rich

deposits in many depositional environments either lacking suitable organic material for radiocarbon dating or too old to be radiocarbon dated. The 30-cm-long OSL samples were inspected under subdued amber light to select the most homogenous section for dating. The outer rim (~1 cm in thickness) and two ends (1-2 cm in length) of a selected core section were cut off and used for water content and dose rate measurements, and the remaining sediments were processed following conventional procedures (Mauz et al., 2002) to extract quartz in particle-size ranges of either 4-11 μm, 75-125 μm, 125-180 μm or 180-

250 μm for equivalent dose ($D_e$) measurement (see also the Supplement). The natural radioactivity of the samples was obtained using a high-resolution, low-level gamma-spectrometer at Tulane University and converted to natural dose rates using conversion factors of Adamiec and Aitken (1998), while the contribution of cosmic radiation was calculated using the formula of Prescott and Hutton (1994). The water content during deposition is assumed to be the same as the measured content. Uncertainty of OSL ages is 3-8% at 1σ and calculated following standard error propagation with uncertainty of

corresponding $D_e$ (2-4% at 1 σ) and natural dose rate (3-8% at 1σ) (Table 1). Thus, variability of OSL-age uncertainty is primarily driven by natural dose rate variability. OSL ages are reported in ka ± 2σ with respect to AD 2010 (Table 1).

The most important requirement for OSL dating is complete bleaching of quartz OSL during the latest sunlight exposure. Water-lain deposit, such as deltaic and beach deposits of this study, may not all be completely bleached because of

attenuation of sunlight spectrum and intensity by turbid water and transport-mode dependent exposure time. Identifying completed bleached deposit relies on (1) making small aliquot or single-grain OSL measurement; (2) using appropriate statistic metrics, and can be aided by using multiple dating methods. In this study, small aliquot measurements were done by mounting sand-sized quartz onto the center 1 to 2 mm diameter area of 10 mm diameter stainless-steel disks. The overdispersion parameter of (Galbraith et al., 1999) and dose distribution are used together for detecting insufficient

bleaching. The statistical procedure of (Arnold et al., 2007) was used to select either a central age model (CAM) or a minimum age model (MAM, see Galbraith et al., 1999) for age calculation for samples measured with sand-sized quartz. A 10% over-dispersion was added in quadrature to the measured $D_e$ error for all aliquots. In addition, experience learned from recent OSL dating in the MPD (Shen and Mauz, 2012; Shen et al., 2015; Shen et al., 2016), OSL ages derived from different grain-size fractions, and radiocarbon ages of this study and from literature are used together to ensure the accuracy of OSL

dating.

Other factors affecting accuracy of OSL dating includes secular disequilibrium in the uranium decay chain and water content variability of the deposit. Recent OSL dating did not find significant secular disequilibrium in the MPD deposits (Shen and Mauz, 2012; Shen et al., 2015; Shen et al., 2016). All OSL samples in this study were taken from near or below local



groundwater level and we applied a 5% uncertainty to water content measured in the lab to account for potential groundwater level variability and long-term compaction of the deposit. The Supplement includes details of the OSL-dating protocol. In total we dated 22 OSL samples

### 3.2.2 Radiocarbon dating

For radiocarbon dating we sliced peat samples into 1 cm segments, sieved them over a mesh of 500 μm and used a microscope to select plant remains for AMS $^{14}$C dating at the University of California, Irvine. If one centimeter did not contain sufficient material, the adjacent centimeter was searched (and so on) until enough material was gathered. The thickest dated interval is 4 cm. For calibration to calendar years we used the IntCal13 curve (Reimer et al., 2013) and OxCal 4.1 (Bronk Ramsey, 2009). In order to facilitate comparison with the OSL ages, the radiocarbon ages are also reported in ka

± 2σ with respect to AD 2010 (Table 2). For the central age, the midpoint of the calibrated 2σ range is used. Since the likelihood of possible ages generally shows a non-normal distribution, this central age may differ slightly from the weighted mean age. In total we dated 22 radiocarbon samples

## 4 New chronology of the Chenier Plain and Mississippi Delta Plain

### 4.1 Chenier Plain

All cross sections in the CP are oriented perpendicular to the chenier of interest. Internally, the cheniers mostly consist of very fine to fine sand with occasionally thick shell hash layers. The front of the chenier is steep, while on the landward side the chenier thins out gradually. All OSL samples taken from the CP, except for sample Creole Ridge I-1, show overdispersion of ~10%, identical to the overdispersion of well-bleached samples from the MDP (Shen et al., 2015). $D_e$ distributions show more than 90% of accepted aliquots falling within the 2σ range of the Central Age Model (CAM) $D_e$

values (Fig. S2), suggesting that the chenier deposits were sufficiently bleached at deposition (cf. Shen and Lang, 2016). Therefore, a CAM was used (Table 1). Creole Ridge I-1 was rejected because it showed ~20% overdispersion that is interpreted as due to post-depositional disturbance and the inclusion of younger grains. OSL ages from individual cheniers are generally in excellent agreement with each other. Some more specific details of the different cross sections are presented below, along with the new chronological data.

### 4.1.1 Little Chenier

Cross section Little Chenier East (LCE, Fig. 3a) shows a gently dipping Pleistocene substrate that is mostly capped by a paleosol and a thin peat bed with ages of 4.0-3.7 ka (Yu et al., 2012). Little Chenier itself is a 2-m-thick sandy deposit with a base around -1 m NAVD. Its front and center contain a prominent shell hash that mainly consists of oyster valves and fragments. The two OSL ages are nearly identical and indicate that this chenier formed 2.9 ± 0.3 ka. Little Chenier West



(LCW, Fig. 3b) exhibits similar dimensions and an age consistent with that from LCE. However, sample LCW V-1 has an age of 2.46 ± 0.20 ka that is regarded as anomalously young with respect to the three OSL ages of ~2.9 ka and was therefore rejected.

### 4.1.2. Chenier Perdue to Pumpkin Ridge

Chenier Perdue has a deep base and an OSL age of 2.6 ± 0.2 ka (Fig. 3c, 4a). The next seaward chenier, Mura Ridge, is dated to 2.20 ± 0.18 ka (Fig. 4a). The most seaward chenier in this cross section, Pumpkin Ridge, is morphologically subdued but it can be traced over a considerable distance. It consists of silt loam or sandy loam with few shell fragments and is dated to 1.66 ± 0.18 ka (Fig. 4a). To the west these three cheniers merge into Creole Ridge (Fig. 2).

### 4.1.3 Grand Chenier (Oak Grove Ridge)

The Grand Chenier paleo-shoreline (Figure 4b) is the most prominent landform of the CP. We dated the portion that is known as the Oak Grove Ridge; the back of the ridge is 1.29 ± 0.10 ka and the front is 1.19 ± 0.12 ka. The base of the chenier is not always easy to pinpoint as it rests on a 2 m thick unit of sandy loam to very fine sand, similar grain sizes as found within the chenier itself. A notable change in relative density and a shift towards slightly darker colored material was used as a marker. The inferred thickness of Grand Chenier is in agreement with the work of Gremillion and Paine (1977)
who studied the stratigraphy of Oak Grove Ridge in detail in three open pits.

### 4.2 Mississippi Delta Plain

All cross sections in the MDP are oriented perpendicular to the main channel of interest. Some more specific details of the different cross sections are presented for each subdelta below, along with the new chronological data. The overdispersion values for OSL samples from the MDP commonly fall between 10-20%, but can be significantly higher for samples younger
than 1 ka (Table 1; cf. Shen et al., 2015). For samples with an overdispersion value <15% more than 90% of aliquots fall within the 2σ band of the selected statistical age model (Fig. S2), suggesting that these samples are not affected by insufficient bleaching. A Minimum Age Model (MAM) and CAM often yield statistically identical ages. The samples with significantly larger overdispersion values are most likely affected by insufficient bleaching and a MAM is used in these cases.

### 4.2.1 Teche subdelta

Cross sections Loreauville and Jeanerette (Figs. 5,6) capture the Teche trunk channel just upstream of the Bayou Cypremort and Bayou Sale bifurcations (Fig. 1b). Both cross sections show a thin peat bed at a depth of -6.5 m NAVD. At three locations the top of the peat bed was dated, yielding nearly identical ages (~6 ka, Table 2). Directly above the peat, unidentified shell fragments are frequently encountered. The coarser sediment body above the peat bed in Loreauville (Fig.
6a) is interpreted as a mouth bar. It is therefore likely that the clay and shells below the mouth bar are part of prodelta



deposits of the Teche subdelta that hence became active in the centuries after 6 ka. The occurrence of reddish clay layers directly above the peat indicate that a portion of the sediment load likely originated from the Red River. In both cross sections the stratigraphy east of Bayou Teche shows two stacked natural-levee deposits separated by flood-basin deposits, indicating two distinct phases of sedimentation. The upper deposits of the older phase are relatively firm due to pedogenesis.

OSL samples from the deeper natural-levee deposits directly adjacent to Bayou Teche have ages of 5.4-4.5 ka. Two OSL samples from the top half of the second phase show ages of 3.7-3.1 ka. The upper sample was derived from a relatively shallow depth within the natural-levee deposits, suggesting that major sedimentation ended here around 3 ka.

Cross sections Donner and Amelia (Fig. 8) still lie along the main channel belt of the Teche subdelta, but downstream of the
10 Bayou Cypremort and Bayou Sale bifurcations (Fig. 1b). Below the peat layer at -2 to -4 m NAVD, natural-levee and flood-basin deposits are present that can be directly linked to the Teche channel belt as they thicken towards it. We dated the base of the peat layer at four sites, but the results cover a wide age range. The youngest age (1.62 ± 0.04 ka) was obtained from site Amelia II where the peat overlies a crevasse-splay deposit. The other samples were taken from peat resting on top of flood-basin deposits and show ages in the range 4.4-2.7 ka. This age discrepancy is partly explained by the relatively high
position of the crevasse-splay deposit in the landscape and hence a lag in peat formation after the abandonment of the Teche subdelta. The large spread is not uncommon and likely reflects a diachronous onset of peat formation in the flood basin after channel abandonment (Törnqvist and Van Dijk, 1993), whereby peat formation commences first in the lowest parts of the flood basin. In such a case the older ages are more representative of the time of abandonment. The spread in ages could, however, also indicate a gradual abandonment of Teche with less widespread sedimentation or a shift to more downstream
sedimentation. It is clear though that sedimentation rates at Donner/Amelia seem to have been very low after ~3.6 ka (sample Donner II-1), since the samples with younger ages (Donner I-2 and Amelia II-2) lie only slightly higher. The top of the peat bed that covers Teche deposits was dated to 1.4-1.2 ka. It underlies flood-basin deposits that thin toward the Teche system and hence we interpret them as originating from the Lafourche system to the east.

### 4.2.2 St. Bernard and Plaquemines-Modern subdeltas

The cross section Burton Road (Fig. 9) shows a peat bed at -4 to -5 m NAVD (Fig. 10a) directly below St. Bernard deposits. Radiocarbon ages from the top of this peat bed show that the St. Bernard subdelta became active shortly after 4 ka (Törnqvist et al., 1996). In the flood basin, the St. Bernard deposits are capped by a peat layer of which the base was dated to 1.4-1.3 ka. Higher up the natural levee a paleosol was formed. Two OSL samples from within the natural-levee deposits return ages of ~2.5 ka and the levee deposits are, hence, older than the overlying peat bed, suggesting that major sedimentation ended well
before peat formation started.

Further downstream, the St. Bernard trunk channel bifurcated into several smaller distributaries. We focused on Bayou Barataria (Fig. 11) as according to Saucier (1963) it was one of the last St. Bernard distributaries to be abandoned. The



western portion of cross section Barataria (Fig. 12) shows natural-levee deposits of Bayou Barataria overlying a silty clay. The stiffness of the clay and the presence of iron oxides within the clay (while the base of the natural-levee deposits lacks iron oxide) indicate subaerial exposure and a time gap. The eastern part of the cross section traverses the inner bend of the channel and shows natural-levee and point-bar deposits. The three OSL ages indicate deposition between 2.6-2.0 ka.

The Plaquemines-Modern system reoccupied the St. Bernard channel (Saucier, 1963) and deposited the sediments above the peat and the paleosol at the Burton Road section (Fig. 10). Two new radiocarbon samples from the top of the peat bed give ages of $1.08 \pm 0.04$ and $0.97 \pm 0.01$ ka, slightly younger than previously published ages. We assume that older ages of the top of this and correlative peat beds (Saucier, 1963; Törnqvist et al., 1996) are more representative of the onset of the

10 Plaquemines-Modern subdelta.

### 4.2.3 Lafourche subdelta

Along the trunk channel that fed the Lafourche subdelta extensive work has been done by Törnqvist et al. (1996) and Shen et al. (2015), showing that its period of activity occurred between 1.6 and 0.6 ka. We focused on the westernmost distributary of the Lafourche subdelta, Bayou du Large (Fig. 8c), as it lies closest to the CP. The stratigraphy is complex with a deep peat

bed at -10 m NAVD overlain by clayey prodelta or bay deposits containing shell hash. Close to the main channel of Bayou du Large this is followed by a natural-levee deposit, further away from the channel belt the deposits become more clayey and organic. The peat bed at -3 to -4 m NAVD separates an older phase of fluvial activity from the most recent one. We dated the top of the peat bead at two sites to $0.9 \pm 0.1$ ka, indicating the start of the last phase of activity of Bayou du Large. This is in agreement with an OSL age of $0.78 \pm 0.10$ ka above the peat. Below the peat bed, the fluvial deposits (Fig. 8) were formed

most likely not too long after the Lafourche subdelta was initiated. Between -2 and -3 m NAVD, reddish-colored sediments indicate a connection between the Lafourche subdelta and the Red River. Red River deposits were also found within the interpreted mouth-bar deposit at -7 m NAVD.

## 5 Paleogeographic evolution

### 5.1 Chenier Plain

Our data show that the Little Chenier paleo-shoreline marks the halt of the Holocene transgression at $2.9 \pm 0.3$ ka. A more landward Holocene shoreline can be excluded, in agreement with the absence of shoreline features landward of Little Chenier (Penland and Suter, 1989). The OSL ages confirm that the CP formed during the past three millennia (Gould and McFarlan, 1959), but have significantly reduced the error margins for the ages of the individual paleo-shorelines.

In order to compare CP evolution with changes in the MDP, we traced the major paleo-shorelines between Calcasieu River and Freshwater Bayou Canal near Vermillion Bay (Fig. 13, Table S2) using previous studies (Russell and Howe, 1935;





Gould and McFarlan, 1959; Penland and Suter, 1989; McBride et al., 2007), digital elevation models (NED 1/3; Gesch, 2007) and Google Earth. Except for the 0.5 ± 0.3 ka paleo-shoreline (Table S2), the chronology is based entirely on the new OSL ages. South of White Lake the reconstructed shoreline positions are the most uncertain since the Grand Chenier paleo-shoreline truncates many older paleo-shoreline features in that area. In most cases a western and eastern segment of a

truncated paleo-shoreline remains and we connected them using the simplest solution.

Using ArcMap we calculated the areas between the paleo-shorelines and divided them by the elapsed time between chenier formation to obtain mass accumulation rates (Figs. 14 and 15), accounting for age uncertainties. Using a constant 2 m thickness of the mudflat sediments (based on Gould and McFarlan, 1959) and a bulk density of 1500 kg m$^{-3}$ we calculated

rates in MT yr$^{-1}$. These are minimum rates as (1) it is unknown how much mudflat erosion may have occurred during chenier formation and (2) it is unknown for how long any given paleo-shoreline remained stationary. If this occurred for a significant amount of time (decades or even centuries) the actual accumulation rates would be higher. Figure 14 shows that between 2.9-1.2 ka mass accumulation rates for the entire CP were fairly constant, fluctuating between 0.5-1 MT yr$^{-1}$. Between the formation of the 1.2 ± 0.1 ka and the 0.5 ± 0.3 ka paleo-shorelines, mass accumulation rate was very high (2.9 ± 1.1 MT yr$^{-1}$,

2σ-range) and during that time about 66% of the current CP area was formed. During the past 0.5 ka the mass accumulation rate for the CP has been slightly negative on average. Local rivers (Calcasieu, Mermentau, Vermillion) transport a negligible 0.13 MT yr$^{-1}$ (Rosen and Xu, 2011) that is probably mostly trapped within the CP.

To study the evolution of different portions of the CP we calculated mass accumulation rates for four coastal segments(Fig.

15). The western (A) and central (B) segments are naturally divided by the Mermentau River. Segment C is the area where a headland was present and segment D is the area east of the headland. All segments show overall growth until ~0.5 ka, except for segment C that faced two periods of significant erosion. Interestingly, the highest rates of accumulation in segment A are not seen after ~1.2 ka as in the other sections, but rather between 1.6-1.2 ka. Erosion of the headland in segment C most likely constituted a significant sediment source to segment A during that time. Overall, the period between 2.5-1.6 ka was

very stable with relatively low accumulation rates and limited erosion of the headland. The shoreline of the CP was straightened considerably during the formation of the prominent Grand Chenier paleo-shoreline around 1.2 ka.

## 5.2 Mississippi Delta Plain

With the new data, the chronology of the Mississippi subdeltas and the paleogeographic evolution of the MDP during the last 6 kyr can be refined (Fig. 14). Activity of the Teche subdelta started sometime after 6.0 ka, the time that a peat bed of that

age was buried by prodelta deposits (Fig. 6). Since by 5 ka a thick natural-levee deposit had formed, it is unlikely that this subdelta was initiated after 5.5 ka (Fig. 14), an interpretation that differs from previous work by Törnqvist et al. (2006). The two stacked natural levees alongside the Teche system (Fig. 6) bracket a period of hardly any activity that may have coincided with the onset of activity of the St. Bernard subdelta shortly after 4 ka (Törnqvist et al., 1996). The end of activity



of the Teche subdelta remains ambiguous, but based on the new data major sedimentation in the study areas seems to have been very limited after 3.5-2.5 ka. This appears to match a period of erosion farther downstream, resulting in a regional ravinement surface (Penland et al., 1988). The prominence of the Teche channel belt on digital elevation maps, suggesting relatively recent activity, is tentatively linked to prolonged occupation of the Teche channel belt by the Red River. This river

currently carries about 4% of the total Mississippi River discharge and formed a smaller pair of natural levees within the much wider alluvial ridge that was created during the peak of activity of the Teche subdelta (Gould and Morgan, 1962). Aslan et al. (2005) put abandonment of the Teche subdelta by the Red River somewhere between 2 and 1 ka, arguing that this was initiated by the progradation of the Lafourche subdelta across Teche distributaries. The Teche channel west of Houma (Fig. 8) was rejuvenated by a Lafourche channel (Gould and Morgan, 1962 and references therein) indicating

complete abandonment of the Teche subdelta by that time. This reconstruction would imply that between 3.5-2.5 ka and the initiation of the Lafourche subdelta, most of the Mississippi River discharge was directed to the St. Bernard subdelta.

The timing for the end of activity of the St. Bernard subdelta is more straightforward, although some uncertainties remain there as well. Along the trunk channel, the base of the peat bed overlying St. Bernard deposits was dated to 1.4-1.3 ka, while

two OSL ages of sandy natural-levee deposits below the peat show ages of 2.6-2.5 ka (Fig. 11). Downstream, along the Barataria distributary, OSL ages indicate activity until at least $2.0 \pm 0.2$ ka. This is close to the initiation of the Lafourche subdelta around 1.7-1.5 ka (based on Törnqvist et al., 1996; Shen et al., 2015). Otvos and Giardino (2004) also report evidence for St. Bernard activity until at least 2 ka. Allowing for some time needed to form the peat bed and the paleosol separating St. Bernard from Plaquemines-Modern deposits, we infer that the St. Bernard subdelta was abandoned before 1.7

20   ka. In this study the top of the dividing peat bed was dated to 1.1-1.0 ka, only slightly younger than the 1.4-1.2 ka age range reported by Törnqvist et al. (1996). This indicates the Plaquemines-Modern subdelta was initiated between 1.4 and 1.0 ka. The end of Lafourche activity was recently dated to 0.6-0.5 ka (Shen et al., 2015).

The most recently formed major distributary is the Atchafalaya River that is depicted as a relatively small channel on maps

from the 16-18[th] centuries. The more detailed maps from the early 19[th] century indicate that the Atchafalaya system was still relatively small at the time (Holland, 2008). Fisk (1952) therefore postulated that only halfway the 19[th] century the Atchafalaya River increased in size and started to deposit significant overbank deposits. This is in agreement with radiocarbon ages of plant material at the base of Atchafalaya overbank strata that fall in the range of 0.20-0.15 ka, with plant material below these deposits dated to 0.6-0.2 ka (Weinstein and Wells, 2004). In the Atchafalaya Bay, sediment directly

below the prodelta deposits of the Wax Lake Delta yielded an OSL age of 0.35-0.30 ka (Shen and Mauz, 2012). It is therefore likely that only after 0.3 ka the Atchafalaya River could have contributed sediment to the longshore current.



## 6 Discussion

### 6.1 Implications for relative sea-level reconstruction from cheniers

Cheniers are erosive geomorphological features that typically form immediately on top of marsh or tidal-flat deposits. The relationship of the elevation of chenier deposits with sea level is not necessarily uniform. For example, Augustinus (1980)

describes two types of cheniers along the shoreline of Surinam: medium to coarse sandy cheniers with a base at the mean high tide level and fine sandy cheniers with a base at the mean low tide level. Anthony (1989) puts the base of cheniers in Sierra Leone between mean sea level and mean spring high tide. Studies from China indicate a base of cheniers near the mean high tide level (Yan et al., 1989; Ying and Xiankun, 1989), while Horne et al. (2015) show cheniers in Australia with a base 0.1-0.2 m above the mean spring low tide level. On the other hand, Dougherty et al. (2012) use the contact between

chenier beach sand and foreshore deposits as a sea-level indicator. In addition, crest elevations of cheniers have been used as a sea-level indicator (McBride et al., 2007). This is problematic though, since their heights may be related to storm-induced wave set-up (Yan et al., 1989; Otvos, 2005) and hence their relationship with sea level is not straightforward. Still, McBride et al. (2007) used average crest heights of cheniers in the CP to reconstruct past sea level, using the average crest height of modern cheniers (~1.2 m NAVD) as an indicator for the relationship between crest heights and sea level. Since the average

crest heights of the cheniers along the Little and Grand Chenier paleo-shoreline are ~2.5 and ~3 m NAVD, respectively, they argued for a higher than present sea level during the formation of these paleo-shorelines. However, using high-resolution sea-level indicators from compaction-free intertidal facies, Yu et al. (2012) showed that RSL was at about -1.5 m NAVD around 3 ka, i.e., during the formation of Little Chenier. This demonstrates that chenier crest heights are not suitable as sea-level indicators.

The cheniers in the CP have undulating bases (Figs. 3 and 4) due to spatially variable erosion patterns; hence chenier bases are problematic sea-level indicators. However, overwash deposits represented by thin sand sheets with a relatively flat base occur landward of the cheniers. Since these deposits formed directly on the pre-existing marsh or tidal flat, we consider the base of these overwash deposits the most suitable sea-level indicator from a chenier. The surface elevation of the marsh

behind the modern chenier is ~0.5 m NAVD on average, just below the highest astronomical tide level for this area. This relationship could be further explored in the future and combined with OSL ages of overwash deposits to reconstruct past sea levels. Here we make the conservative assumption that these features define the maximum elevation of mean sea level during chenier formation and obtain upper limiting data points from the base of overwash deposits using the protocol outlined in Hijma et al. (2015, Table S4). The new data fill the gap that existed in the Holocene RSL synthesis for the CP

and MDP (Yu et al., 2012) and show that sea level remained below present mean sea level in the CP during the late Holocene (Fig. 16).



## 6.2 Coupled Mississippi Delta Plain-Chenier Plain evolution

Figure 14 shows that the progradation history of the CP is dominated by one major episode, namely the period between 1.2 and 0.6 ka when a westward thinning wedge of sediment accumulated that forms 66% of the current CP area. The thinning pattern is distinct, exemplified by relatively low accumulation rates in the most westward segment (Figure 15), pointing towards a sediment source east of the CP, i.e. the MDP. Since the timing of this episode corresponds closely with the period of activity of the Lafourche subdelta, we hold the shift from the St. Bernard to the Lafourche subdelta responsible for this period of rapid progradation. Prior to this period, progradation rates were rather constant, while after this period the CP was relatively stable with increased erosion in recent times, likely due to recent accelerated sea-level rise and sediment starvation. We agree with McBride et al. (2007) that especially near the CP river mouths local effects resulted in deviations from this general picture of CP evolution, resulting in spits and curved beach ridges.

The individual evolution of the four segments, however, also shows marked differences that require further explanation. An important feature during CP evolution was the headland south of White Lake that is linked to the buried deposits of the Lafayette meander belt of the ancestral Mississippi River (Fig. 1a). This headland was especially prominent between 2.9-2.5 ka, but was in place until the paleo-shoreline was straightened at 1.2 ka. West of the headland a bay was present, bounded to the west by the Calcasieu River mouth (Fig. 13). We argue that the infill of this bay was to a large extent fed by headland erosion and the resulting abundant sediment. This is illustrated by the match of two distinct phases of headland erosion with two equally distinct phases of accumulation in segment A. We rule out the possibility that the infill of the bay was dominated by sediment from contemporary Mississippi subdeltas as accumulation rates in segment D, closest to the MDP, were not particularly high and much lower than during Lafourche activity. Building upon the notion that segment D is the most sensitive to changes in the position of the main Mississippi River mouth and accepting that the Lafourche subdelta sediment output was responsible for overall rapid progradation between 1.2-0.6 ka, we argue that between 2.9 ka and the initiation of the Lafourche subdelta (1.7-1.5 ka) the locus of Mississippi sediment output was situated east of the Lafourche subdelta. In other words, during roughly the first half of CP evolution, the St. Bernard subdelta carried most of the discharge, in agreement with Figure 14. If the Teche subdelta was still active to a significant extent, this should have resulted in more rapid accumulation rates than what is recorded, especially since the Teche subdelta lies closer to the CP than the Lafourche subdelta.

The question that then arises is what caused CP progradation to start around 2.9 ka. Fisk (1948) and Penland and Suter (1989) hypothesized that this was due to Teche and Lafourche activity, respectively, which is untenable in view of the new chronological data. Gould and McFarlan (1959) linked the change to the initiation of Bayou Barataria, the most western distributary of the St. Bernard subdelta. Our new data indicate that Bayou Barataria was indeed active during that time and could have contributed sediment to the longshore current. In addition to this, the strong erosion of the Teche subdelta



promontory (see e.g. Penland et al., 1987)  most likely occurred during this timeframe as well and would have formed a substantial sediment source. However, these two sediment sources cannot explain the shift from overall transgression to overall progradation around 2.9 ka, since the close proximity of the Teche subdelta and its activity in the millennia before 2.9 ka would have resulted in an equally or most likely even larger sediment source. We therefore argue that the shift to

5      overall progradation was triggered by a gradual slowdown in the rate of RSL rise (Figure 16) in combination with abundant local sediment supply from the eroding headland, possibly augmented by the eroding Teche subdelta. The ~1.5 m rise of RSL during the past 3 kyr (Figure 16) is driven by regional subsidence, mainly due to glacial isostatic adjustment (Yu et al., 2012). Sea-level oscillations on the order of a few decimetres have been proposed for this time period (e.g. González and Törnqvist, 2009) and may have had an, at this point undetermined, impact on CP evolution.

The above explains the start of the CP formation around 2.9 ka and the period of rapid accumulation after 1.2 ka, but several issues remain. The first concerns the ~20 km westward drift of the mouth of the Mermentau River in the past 1.6 kyr. We tentatively link this to the final infill of the bay of the Mermentau River. Around 2.5 ka the paleo-shoreline reconstruction (Fig. 13) still shows the presence of a bay, while the 1.6 ka paleo-shoreline is much straighter. This would have resulted in a

stronger influence of the longshore current on river-mouth morphology and the formation of spits that forced the river mouth to shift westward. This was aided by abundant sediment supply of the eroding headland between 1.6-1.3 ka and the Lafourche subdelta. A second issue concerns the formation of the very prominent and wide Grand Chenier paleo-shoreline that straightened the shoreline of the CP, hereby causing renewed erosion of the headland that had been rather stable for nearly 1 kyr. Our OSL ages indicate that Grand Chenier formation started around 1.3 ka and lasted for at least a century (Fig.

4b). The large width, in comparison to the other cheniers, and the fact that the OSL ages decrease in a seaward direction are indicative of progradation. Both Penland and Suter (1989) and McBride et al. (2007) linked the formation of the Grand Chenier paleo-shoreline to changes within the Lafourche subdelta, hence suggesting that the main river mouth of the subdelta shifted east. At present this cannot be substantiated with chronological data from the Lafourche subdelta. In addition, McBride et al. (2007) suggested RSL rise as an important factor in the formation of the Grand Chenier paleo-

shoreline. Data from González and Törnqvist (2009) indeed do suggest relatively high rates of RSL rise between 1.2-0.8 ka, so within the range of Grand Chenier formation. Figure 14 allows for a third explanation, namely that the formation of the Grand Chenier paleo-shoreline is linked to the initiation of the Plaquemines-Modern subdelta.

**6.3 Implications for coastal restoration**

Within Louisiana's Coastal Master Plan (CPRA, 2017) to battle land loss due to RSL rise and sediment deficits, $5 billion

30     has been dedicated to sediment diversions. The intent is to lose less sediment to the Gulf of Mexico and instead use the sediment to create new land within the MDP. This is also the focus of *ChangingCourse.us*, an independent initiative that has solicited plans to restore the natural land-building capacity of the river while maintaining the navigation system. The potential of creating new land using Mississippi River sediment was demonstrated during the 2011 flood (Allison et al.,



2012; Nittrouer et al., 2012). In some of the plans sediment of the Atchafalaya River is diverted to the Terrebonne Marshes, while further east diversions have been proposed to Barataria Bay and Breton Sound (Fig. 1). The current focus lies on these two latter locations.

5   It can be expected that due to diversions the delivery of Mississippi River sediment to the longshore current will change. However, most of the sediment will initially be trapped within MDP bays and hence will not reach the CP. This is currently also the case for the Atchafalaya River, of which only 0.5% of the transported 70 MT yr$^{-1}$ reaches the CP (Draut et al., 2005b), although this is still sufficient to cause progradation in the eastern CP. This percentage is strikingly similar to what we have reconstructed for the active phase of the Lafourche subdelta when $2.9 \pm 1.1$ MT yr$^{-1}$ accumulated in the CP.

10 Assuming that the Mississippi River had a sediment load somewhere between 200-400 MT yr$^{-1}$, this means that about 0.5-1.5% of the sediment ended up in the CP. This indicates that the planned diversions have the potential to also result in a slowdown of CP erosion, especially after some of the MDP bays have been filled in. In summary, our results show that if only 0.5-1.5% of the total Mississippi River sediment load would reach the CP erosion rates can be expected to decrease considerably, although this effect will be partly counteracted by the projected increase in the rate of RSL rise.

15 **7 Conclusions**

This paper shows that the evolution of the Mississippi Delta Plain (MDP) and the adjacent Chenier Plain (CP) is interlinked. Based on OSL and radiocarbon dating we conclude that the CP started to form around 3 ka. Large-scale patterns in the evolution of the CP are a direct consequence of shifting subdeltas, in addition to changes in regional sediment sources and rates of RSL change. We use the base of chenier-overwash deposits to show that RSL rose steadily during the past 3 kyr;

20 contrary to what has been suggested before it never reached an elevation higher than present.

The period with the highest accumulation rates in the CP (1.2-0.6 ka) is directly linked to a westward shift of the Mississippi River, resulting in abundant sediment supply. The $2.9 \pm 1.1$ MT that accumulated each year on the CP during this period corresponds to 0.5-1.5% of the total sediment load of the present-day Mississippi River. Remarkably, roughly the same

25 percentage of the Atchafalaya sediment load is currently reaching the CP and resulting in local shoreline progradation. This suggests that if proposed Mississippi River sediment diversions are planned carefully, a slowdown of erosion can be expected not only in and near the new subdelta, but also along the shoreline of the CP. The information on the interlinked CP-MDP evolution in this paper could be used to constrain future generations of numerical models to obtain more robust predictions of the effects of sediment diversions on the evolution of both the MDP and the CP. A marked difference with the

30 past, however, is that the CP evolved under conditions of relatively slow rates of RSL rise. It therefore remains to be seen whether the CP can survive the currently ongoing acceleration of sea-level rise, even if sediment supply increases.



**Author contribution**

M.P.H. and T.E.T. designed the project. M.P.H. led all the fieldwork and prepared the radiocarbon samples. Z.S. was involved in fieldwork and, together with B.M., prepared, dated and analysed all OSL samples. M.P.H. composed the manuscript with major input from Z.S. and T.E.T. All figures, except for Figure 15 (Z.S.), were created by M.P.H.

5 **Competing interests**

The authors declare that they have no conflict of interest.

**Acknowledgements**

We would like to thank Jennifer Kuykendall, Arielle Woods, Elizabeth Chamberlain, Krista Jankowski and Jon Marshak for field assistance. Arielle Woods also assisted with reconstructing chenier paleo-shorelines. We acknowledge all the
10 landowners for allowing us to drill on their land. We are grateful to the Rockefeller Wildlife Refuge for their hospitality during our stay in the Chenier Plain. The fieldwork along Bayou Barataria was possible due to the kind assistance of Julie Whitbeck of the National Park Service who allowed us to drill in the Barataria Preserve of the Jean Lafitte Natural Historic Park and Preserve. Lee Newsom (Penn State) helped with identifying macrofossils for radiocarbon dating. Funding was provided by the US Department of Energy through the National Institute for Climatic Change Research Coastal Center. This
15 is a contribution to the PALSEA programme.





Figure 1. Digital elevation maps (NED 1/1 arc, Gesch, 2007; LSU, 2011) of the study areas, including (a) the Chenier Plain with the main cheniers indicated by the black lines; and (b) the Mississippi Delta Plain with the position of its subdeltas. The outline of the subdeltas is essentially the same as in Frazier (1967), but in line with Fisk (1944) the Teche subdelta extends farther east. BR-Back Ridge; MSR-Mesquitte Ridge; TI-Tiger Island; LPI-Little Pecan Island; NI-North Island; PI(BR)-Pecan Island (Back Ridge); MI-Mulberry Island; KR-Kochs Ridge; BI-Belle Island; CAT-Chenier au Tigre. In the updated-chronology box the bold numbers indicate the period of activity, while the smaller numbers in italic reflect the possible period of activity (see also Fig. 14).





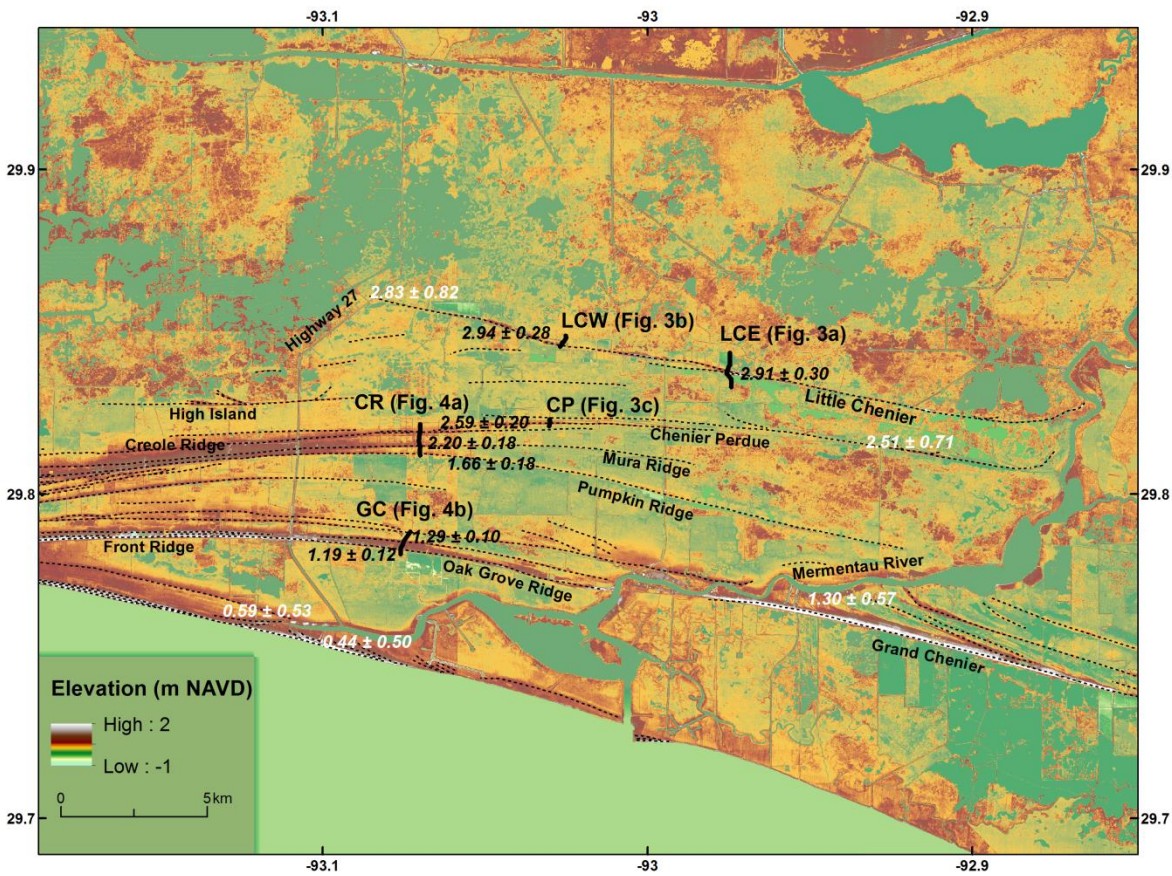

Figure 2. Digital elevation map (NED 1/3 arc) of the Chenier Plain study area (for location see Fig. 1a) with the location of the different cross sections: Little Chenier East (LCE), Little Chenier West (LCW), Chenier Perdue (CP), Creole Ridge (CR, consisting of Chenier Perdue, Mura Ridge and Pumpkin Ridge) and Grand Chenier/Oak Grove Ridge (GC). The OSL ages (Table 1) are shown in black, selected radiocarbon ages from Gould and McFarlan (1959) are in white (Table S1). The OSL ages and the calibrated radiocarbon ages are expressed in ka (± 2☐) with respect to AD 2010. The dotted lines indicate the position of cheniers.





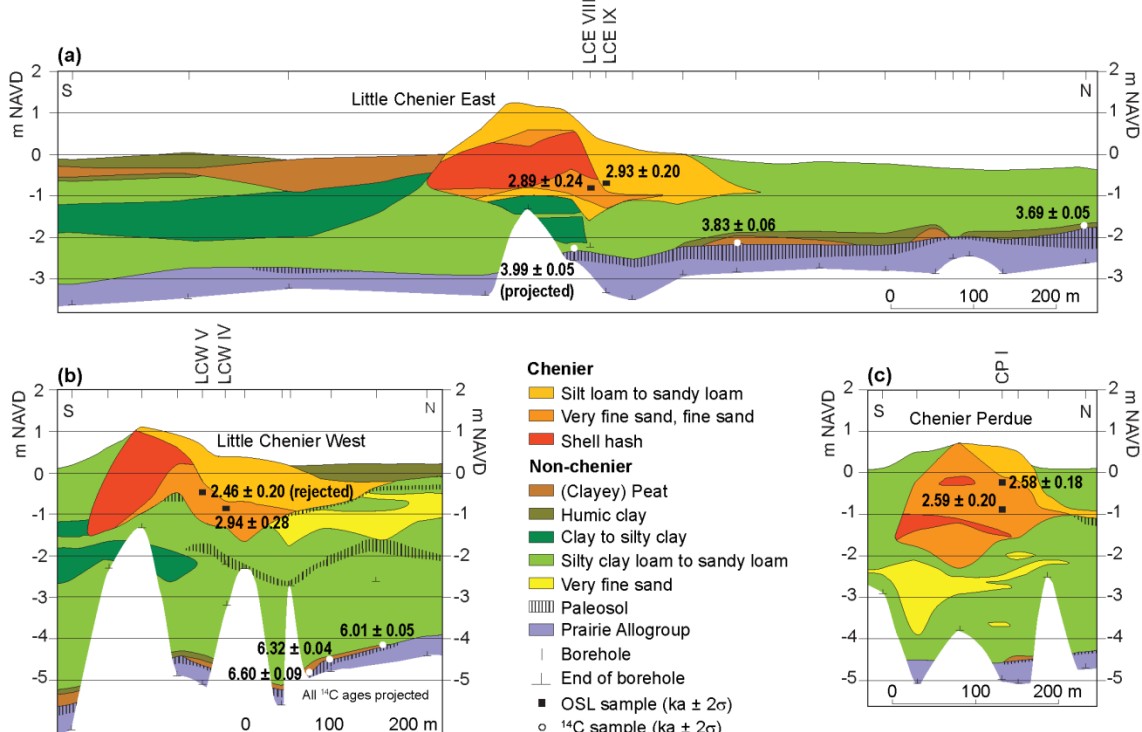

Figure 3. Cross sections across (a) Little Chenier East, (b) Little Chenier West and (c) Chenier Perdue and the stratigraphic position of the OSL samples (Table 1). For location of cross sections see Fig. 2. The radiocarbon ages are from Yu et al. (2012).





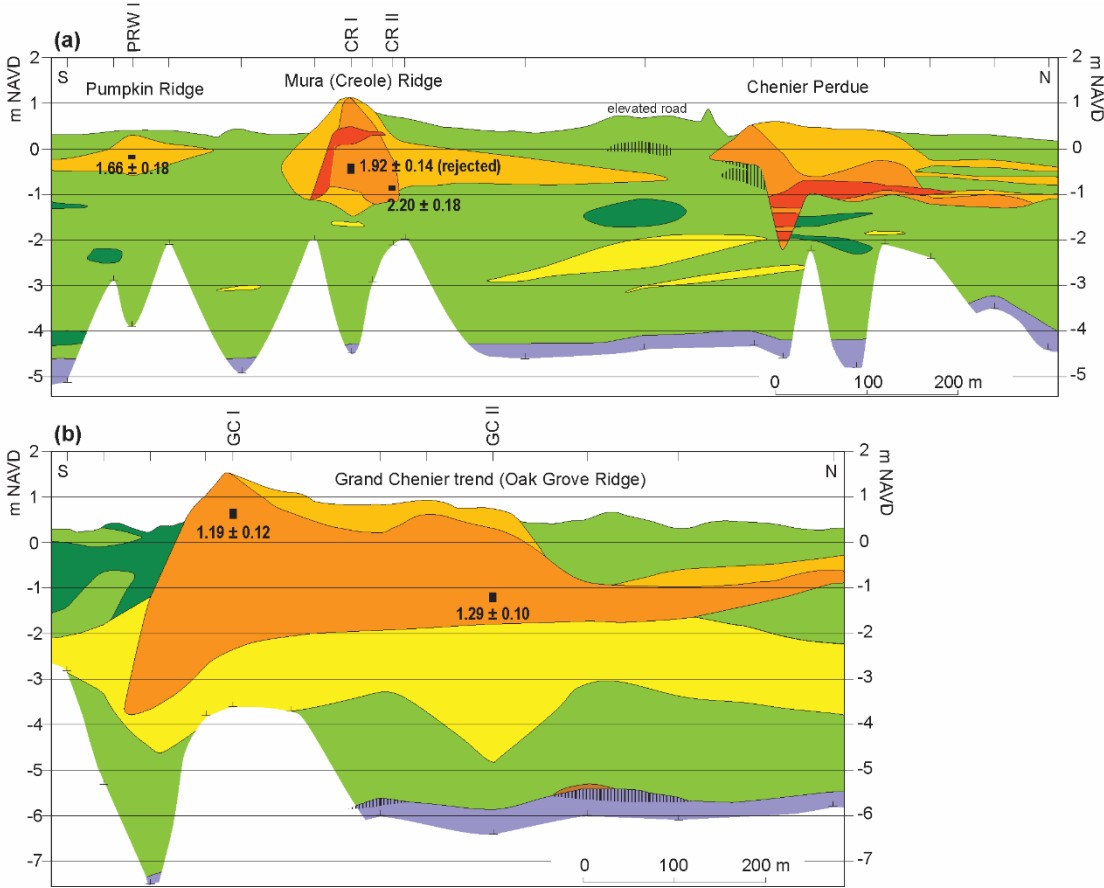

Figure 4. Cross sections across (a) Chenier Perdue-Creole Ridge-Pumpkin Ridge and (b) Grand Chenier (Oak Grove Ridge) and the stratigraphic position of the OSL samples (Table 1). For location of cross sections see Fig. 2; see Fig. 3 for legend. Sample CR I-1 showed ~20% overdispersion and was therefore rejected.





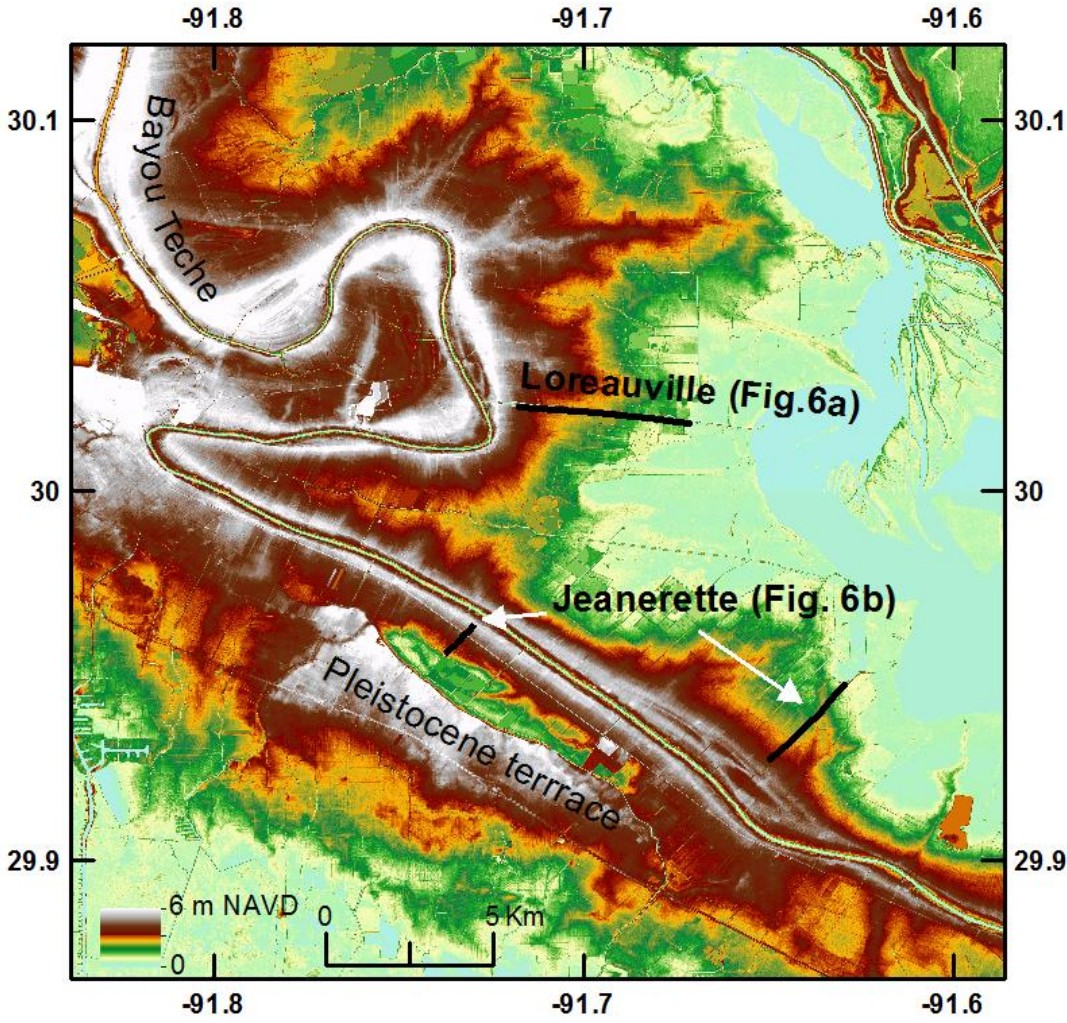

Figure 5. Digital elevation map (NED 1/3 arc) of the Bayou Teche system near New Iberia (for location see Fig. 1b) with the location of cross sections Loreauville and Jeanerette (Fig. 6).





Figure 6. Cross sections (a) Loreauville and (b) Jeanerette with the stratigraphic position of the OSL (Table 1) and radiocarbon samples (Table 2).





Figure 7. Digital elevation map (NED 1/3 arc) of Bayou Teche and Bayou du Large near Houma (for location see Fig. 1a) with the location of cross sections Amelia, Donner and Theriot (Fig. 8).





Figure 8. Cross sections (a) Amelia, (b) Donner and (c) Theriot with the stratigraphic position of the OSL (Table 1) and radiocarbon samples (Table 2). See Figure 6 for legend.





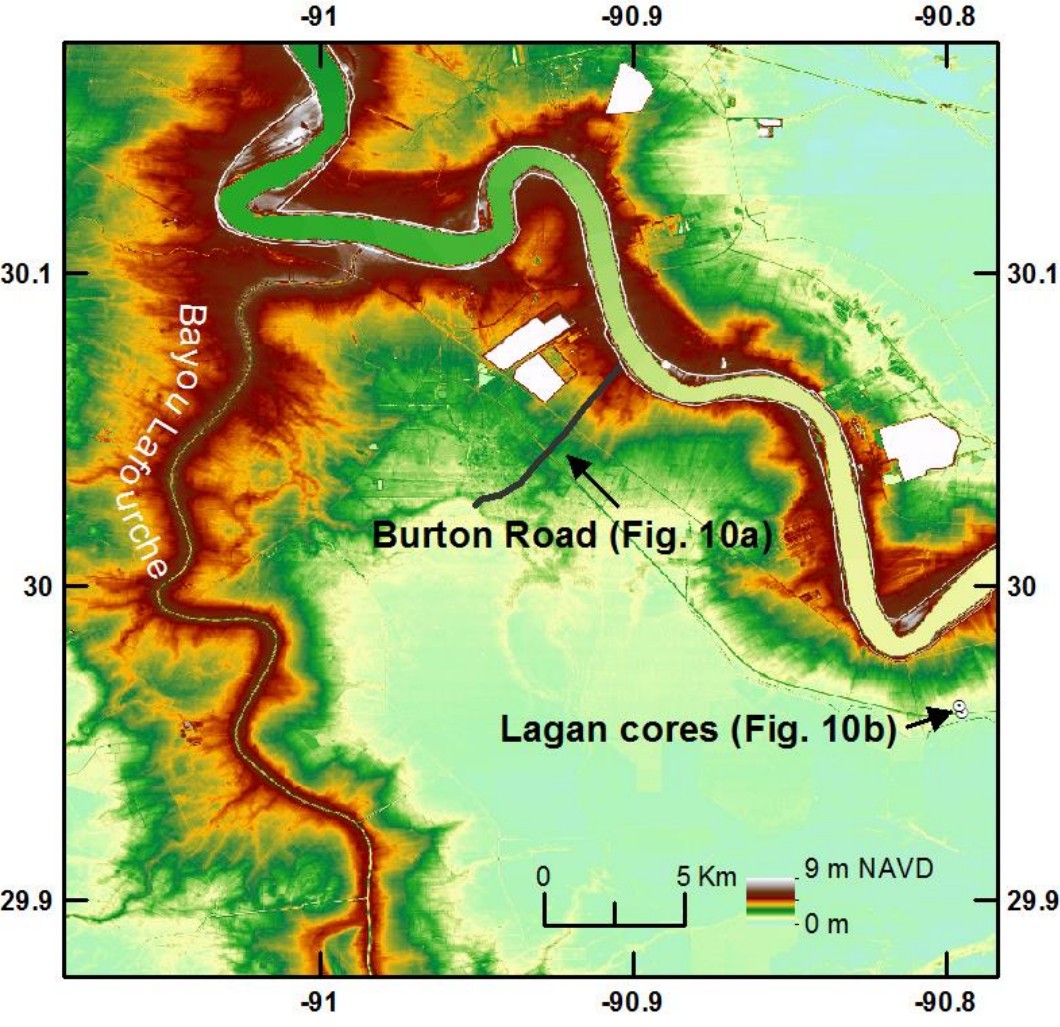

Figure 9. Digital elevation map (NED 1/3 arc) of the modern Mississippi River downstream of the Bayou Lafourche bifurcation (for location see Fig. 1b) with the location of cross section Burton Road and the Lagan cores (Fig. 10).




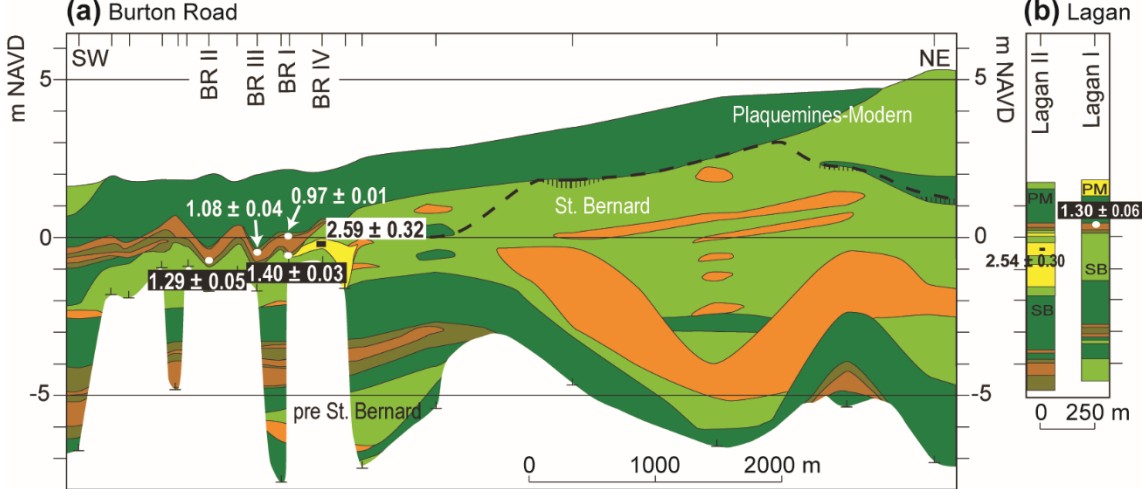

Figure 10. Cross section (a) Burton Road and the cores at (b) Lagan with the stratigraphic position of the OSL (Table 1) and radiocarbon samples (Table 2). The radiocarbon age from Lagan I is from Törnqvist et al. (1996). See Figure 6 for legend. PM=Plaquemines-Modern; SB=St. Bernard.



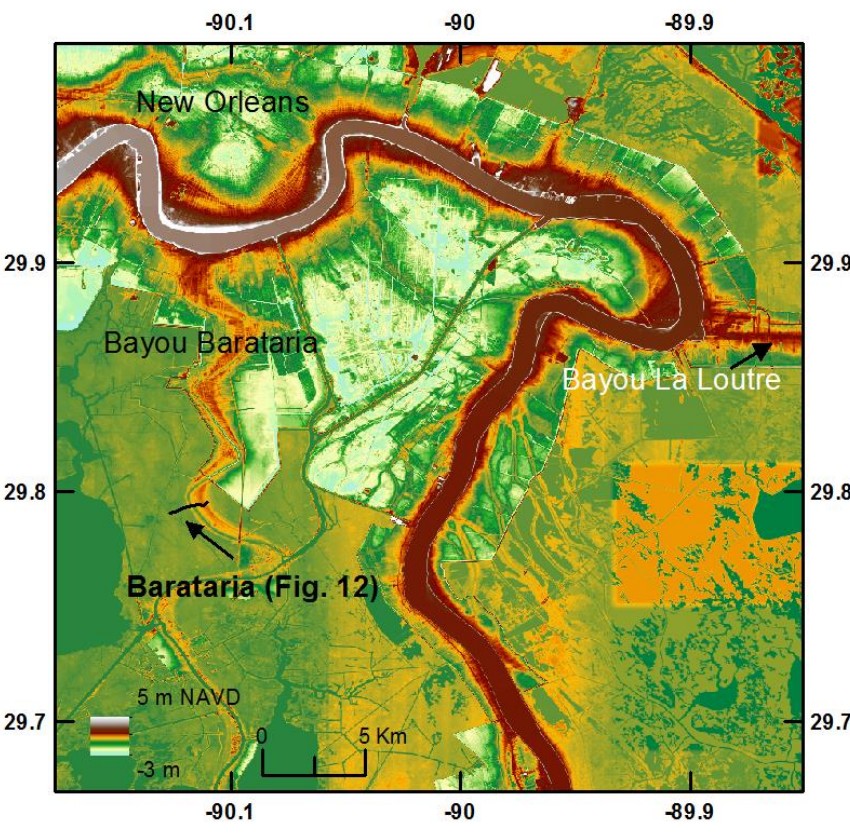

Figure 11. Digital elevation map (NED 1/3 arc) of the Barataria area (for location see Fig. 1b) with the location of cross section Barataria (Fig. 12).





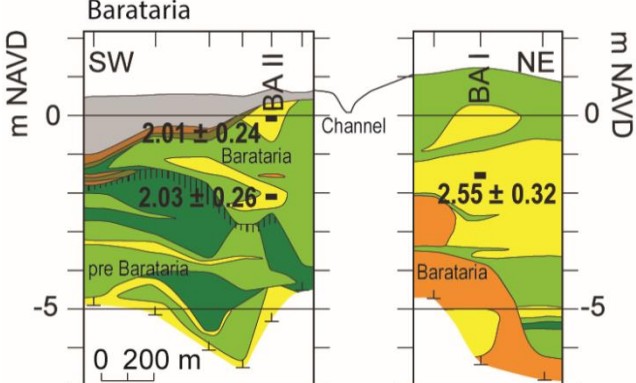

Figure 12. Cross section Barataria with the stratigraphic position of the OSL samples (Table 1). See Figure 6 for legend.

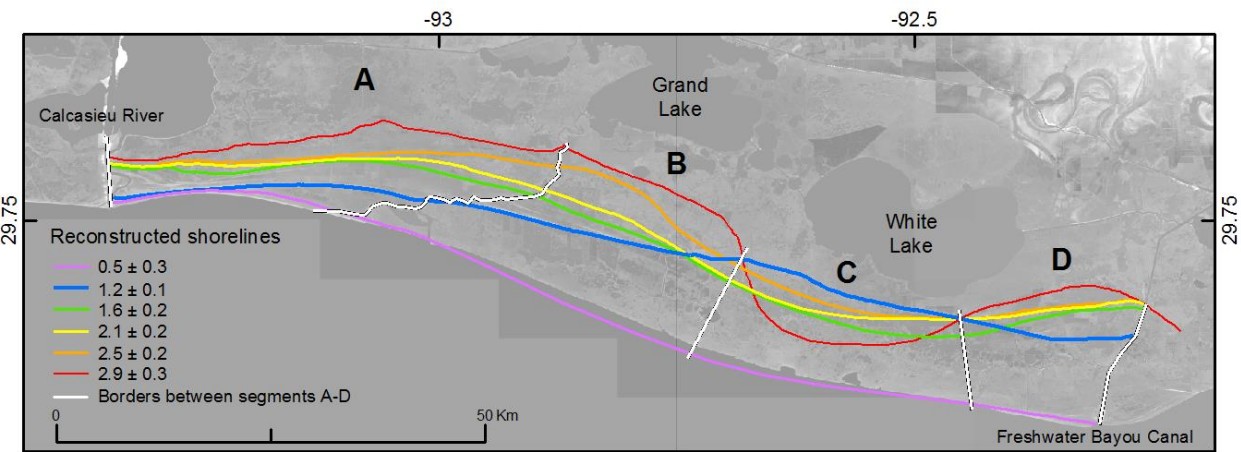

Figure 13. Six paleo-shorelines reconstructed from the Chenier Plain, along with the location of coastal segments A-D. See

5    Table S2 for background information on the used chronology.





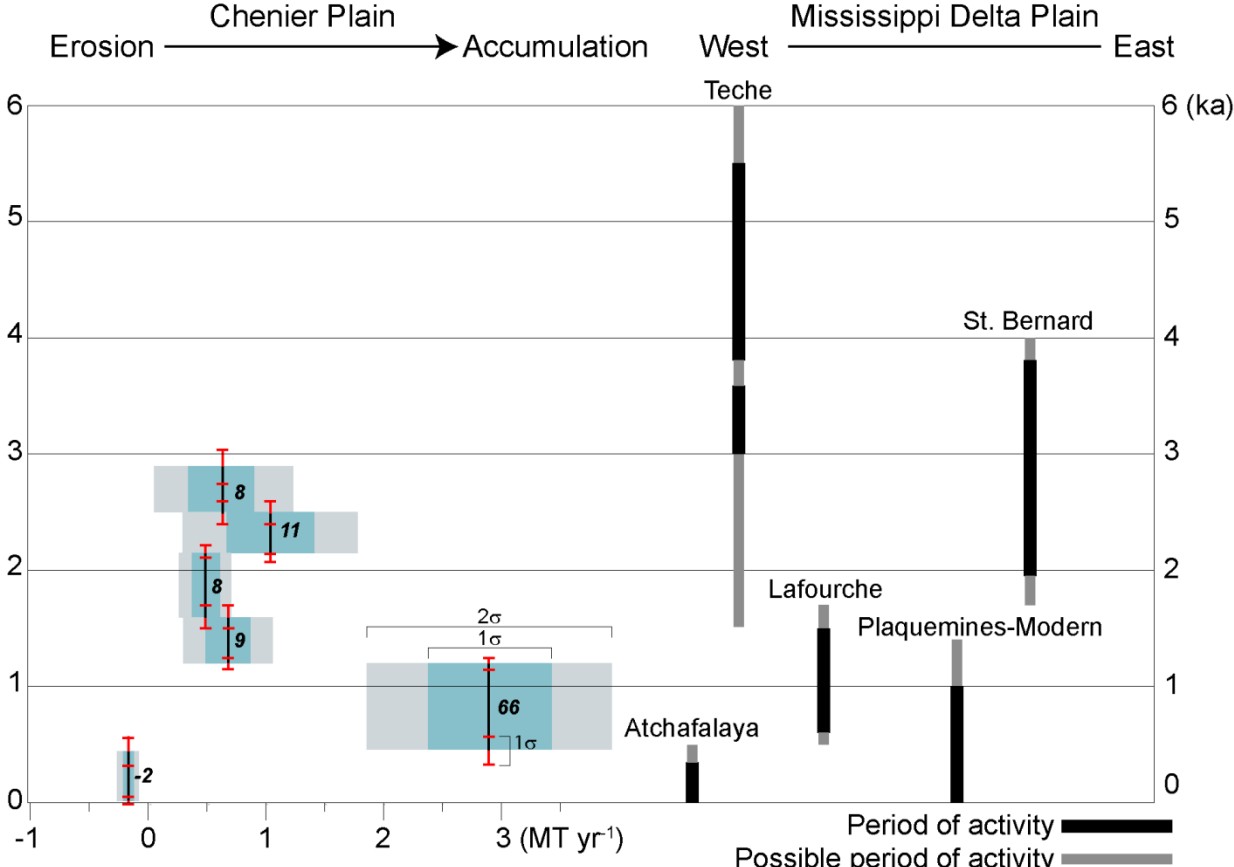

Figure 14. Accumulation patterns for the Chenier Plain and the chronology and the relative position of subdeltas in the Mississippi Delta Plain during the past 6 ka. The numbers next to the vertical error bar of the accumulation rates show the relative contribution to the total accumulation for each period of accumulation. The vertical error bars are derived from the inferred ages in Table S2, while the horizontal error bars account for the uncertainty in the accumulate rate due to ages uncertainties.





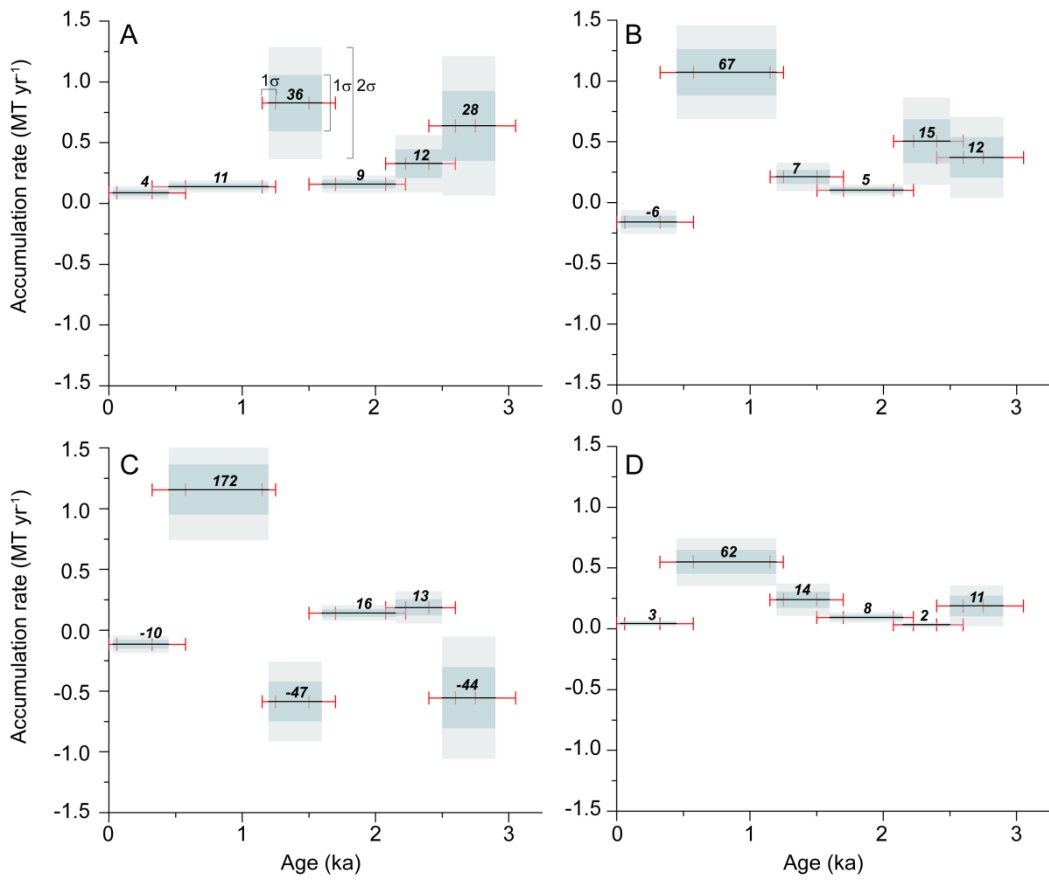

5    Figure 15. Mass accumulation rates for coastal segments A-D in the Chenier Plain (Fig. 13). Since the segments have

different sizes, the relative contribution to the total accumulation in each segment was calculated to facilitate comparison.

They are plotted above to the horizontal error bar.





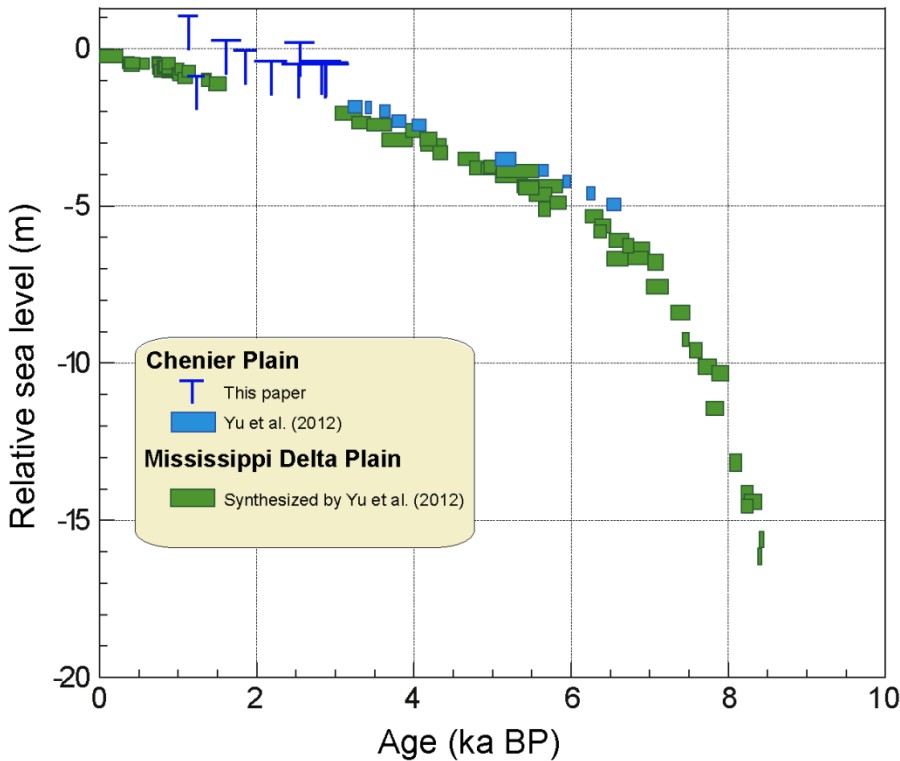

Figure 16. Comparison of Holocene RSL records derived from cheniers (this study) and basal peat from the Chenier Plain and the Mississippi Delta Plain (Yu et al., 2012). The chenier-overwash data are interpreted as upper limiting data (see text). For all limiting data, the width of the horizontal bar is defined by the 2σ age error and the length of the vertical bar by its 2☐

5  error range (Table S4).

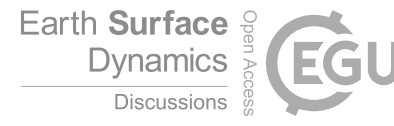



**Table 1. Details of OSL data.**

| Sample name | Lab code | Latitude† | Longitude | Z† (m) | Depth below surface (m) | $^{238}$U±1σ (µg g⁻¹) | $^{232}$Th±1σ (µg g⁻¹) | $K_2O$±1σ (µg g⁻¹) | Water content | $d_{cosmic}$ (Gy kyr⁻¹) | Grain size (µm) | na‡ | Over-dispersion | Age model | De±1σ (Gy) | OSL age ±1σ (ka before AD 2010) |
|---|---|---|---|---|---|---|---|---|---|---|---|---|---|---|---|---|
| **Chenier Plain** | | | | | | | | | | | | | | | | |
| Chenier Perdue I-1 | LV458 | -93.0340 | 29.8223 | 0.6 | 0.73-0.85 | 0.84±0.03 | 2.41±0.14 | 1.05±0.03 | 0.18±0.05 | 0.20 | 125-180 | 46/68 | 10±2% | CAM | 3.58±0.07 | 2.58±0.09 |
| Chenier Perdue I-2 | LV459 | -93.0340 | 29.8223 | 0.6 | 1.42-1.55 | 0.57±0.02 | 2.09±0.11 | 0.98±0.03 | 0.17±0.05 | 0.18 | 125-180 | 34/48 | 8±2% | CAM | 3.14±0.07 | 2.59±0.10 |
| *Creole Ridge I-1** | LV465 | -93.0740 | 29.8149 | 1.1 | 1.45-1.65 | 0.95±0.03 | 2.75±0.11 | 0.96±0.03 | 0.20±0.05 | 0.18 | 125-180 | 52/72 | 19±2% | CAM | 2.42±0.06 | 1.92±0.07 |
| Creole Ridge II-1 | LV593 | -93.0743 | 29.8152 | 0.7 | 1.46-1.57 | 0.67±0.02 | 2.16±0.07 | 1.16±0.03 | 0.20±0.05 | 0.18 | 180-250 | 15/48 | 3±4% | CAM | 2.92±0.07 | 2.20±0.09 |
| Grand Chenier I-1 | LV462 | -93.0800 | 29.7832 | 1.5 | 0.75-0.94 | 1.72±0.04 | 5.99±0.15 | 1.33±0.04 | 0.15±0.05 | 0.20 | 125-180 | 22/48 | 8±2% | CAM | 2.39±0.05 | 1.19±0.06 |
| Grand Chenier II-1 | LV463 | -93.0790 | 29.7857 | 0.7 | 1.87-2.07 | 1.40±0.04 | 4.62±0.18 | 1.25±0.04 | 0.20±0.05 | 0.16 | 125-180 | 41/72 | 9±2% | CAM | 2.15±0.04 | 1.29±0.05 |
| Little Chenier West IV-1 | LV582 | -93.0300 | 29.8470 | 0.4 | 1.20-1.30 | 2.03±0.05 | 6.46±0.19 | 1.24±0.04 | 0.22±0.05 | 0.19 | 75-125 | 20/48 | 8±3% | CAM | 5.51±0.16 | 2.94±0.14 |
| *Little Chenier West V-1** | LV460 | -93.0302 | 29.8468 | 0.6 | 0.95-1.06 | 1.82±0.06 | 6.11±0.21 | 1.36±0.04 | 0.21±0.05 | 0.19 | 125-180 | 56/72 | 9±1% | CAM | 4.68±0.07 | 2.46±0.10 |
| Little Chenier East VIII-1 | LV461 | -92.9793 | 29.8384 | 0.4 | 1.11-1.22 | 2.22±0.06 | 6.54±0.21 | 1.38±0.04 | 0.23±0.05 | 0.19 | 125-180 | 44/48 | 10±2% | CAM | 5.48±0.10 | 2.89±0.12 |
| Little Chenier East IX-1 | LV583 | -92.9793 | 29.8386 | 0.3 | 1.12-1.22 | 1.86±0.05 | 6.02±0.20 | 1.39±0.04 | 0.22±0.05 | 0.19 | 75-125 | 14/48 | 6±4% | CAM | 5.69±0.19 | 2.93±0.15 |
| Pumpkin Ridge West I-1 | LV464 | -93.0741 | 29.8128 | 0.4 | 0.49-0.55 | 2.55±0.07 | 8.80±0.25 | 1.66±0.05 | 0.21±0.05 | 0.21 | 75-125 | 34/48 | 6±2% | CAM | 4.10±0.07 | 1.66±0.09 |
| **Mississippi Delta Plain** | | | | | | | | | | | | | | | | |
| Barataria I-1 | LV592 | -90.1160 | 29.7960 | 1.2 | 2.71-2.85 | 2.98±0.07 | 8.76±0.18 | 1.90±0.04 | 0.28±0.05 | 0.14 | 75-125 | 15/24 | 14±3% | MAM | 6.12±0.24 | 2.55±0.16 |
| Barataria II-1 | LV590 | -90.1241 | 29.7949 | 0.7 | 0.73-0.85 | 3.46±0.08 | 10.35±0.20 | 1.93±0.04 | 0.27±0.05 | 0.20 | 75-125 | 18/48 | 12±3% | CAM | 5.37±0.18 | 2.01±0.12 |
| Barataria II-2 | LV591 | -90.1241 | 29.7949 | 0.7 | 2.73-2.85 | 4.29±0.10 | 11.00±0.21 | 2.11±0.05 | 0.31±0.05 | 0.14 | 75-125 | 23/48 | 8±2% | CAM | 5.60±0.13 | 2.03±0.13 |
| Burton Road IV-1 | LV589 | -90.9387 | 30.0367 | 2 | 2.20-2.35 | 3.59±0.09 | 9.76±0.18 | 2.02±0.05 | 0.27±0.05 | 0.15 | 75-125 | 54/144 | 17±2% | MAM | 6.98±0.18 | 2.59±0.16 |
| Jeanerette I-1 | LV466 | -91.7363 | 29.9610 | 4.4 | 7.40-7.70 | 3.26±0.09 | 8.56±0.33 | 2.13±0.07 | 0.21±0.05 | 0.06 | 75-125 | 49/60 | 16±2% | MAM | 12.44±0.26 | 4.46±0.37 |
| Jeanerette II-1 | LV467 | -91.7350 | 29.9623 | 4.9 | 5.30-5.60 | 3.48±0.08 | 9.31±0.26 | 1.85±0.05 | 0.26±0.05 | 0.09 | 75-125 | 21/48 | 9±3% | CAM | 12.6±0.3 | 5.06±0.30 |
| Jeanerette II-2 | LV468 | -91.7350 | 29.9623 | 4.9 | 5.70-6.00 | 3.55±0.08 | 9.82±0.26 | 1.79±0.05 | 0.27±0.05 | 0.08 | 75-125 | 20/48 | 10±3% | CAM | 13.3±0.4 | 5.42±0.33 |
| *Weighted mean age* | | | | | | | | | | 4-11 | | 8/8 | 2±2% | | 15.05±0.27 | 5.41±0.24 |
| Jeanerette IV-1 | LV594 | -91.6441 | 29.9361 | 2.5 | 0.65-0.78 | 3.66±0.09 | 10.43±0.18 | 1.67±0.04 | 0.20±0.05 | 0.2 | 75-125 | 28/96 | 20±3% | CAM | 8.72±0.35 | 3.10±0.21 |
| Jeanerette IV-2 | LV595 | -91.6441 | 29.9361 | 2.5 | 2.05-2.15 | 2.80±0.07 | 9.75±0.19 | 2.08±0.05 | 0.25±0.05 | 0.16 | 75-125 | 17/24 | 10±2% | CAM | 9.85±0.28 | 3.67±0.24 |
| Lagan II-1 | LV588 | -90.7985 | 29.9614 | 1.7 | 2.00-2.15 | 3.91±0.09 | 11.45±0.21 | 1.76±0.04 | 0.34±0.05 | 0.16 | 75-125 | 34/96 | 16±3% | MAM | 6.09±0.24 | 2.54±0.15 |
| Theriot III-1 | LV586 | -90.7395 | 29.4804 | 0.9 | 1.85-1.95 | 3.69±0.09 | 10.82±0.21 | 1.92±0.04 | 0.25±0.05 | 0.17 | 75-125 | 62/88 | 40±4% | MAM | 2.16±0.06 | 0.78±0.05 |

*Rejected dates, they are considered anomalously young. *Coordinates were measured with a handheld GPS and rounded to 4 decimals. Coordinates of the Lagan site were previously obtained from a map (Törnqvist et al., 1996). † All elevations are relative to NAVD 88 and obtained from the National Elevation Dataset 1/3 Arc-second from the USGS (vertical error of ~0.25 m). ‡ na=number of accepted aliquots/number of measured aliquots.



**Table 2. Details of radiocarbon data.**

| Sample name | Lab code (UCIAMS) | X† | Y | Z‡ (m) | Depth below surface (m) | Material dated | Age (¹⁴C yr BP) | Mean age ± 2σ (cal ka before AD 2010) |
|---|---|---|---|---|---|---|---|---|
| Amelia I-1* | 107354 | -91.0396 | 29.7422 | 0.8 | 0.83-0.84 | 3 *Nyssa aquatica* stone fragments | 2180±25 | 2.27±0.09 |
| Amelia I-1b | 109238 | -91.0396 | 29.7422 | 0.8 | 0.82-0.85 | 1 *Rhynchospora* sp. fruit, 2 *Scirpus* spp. achenes | -240±15 | Modern |
| Amelia I-2 | 107355 | -91.0396 | 29.7422 | 0.8 | 0.95-0.97 | 1 *Rosa palustris* thorn, 2 unidentified leaf fragments | -845±15 | Modern |
| Amelia II-1 | 107352 | -91.0319 | 29.7479 | 0.9 | 2.57-2.59 | 1 *Nyssa aquatica* stone | 1200±15 | 1.18±0.05 |
| Amelia II-2 | 107353 | -91.0319 | 29.7479 | 0.9 | 3.53-3.57 | 8 *Taxodium distichum* bark fragments | 1645±15 | 1.62±0.04 |
| Burton Road I-1 | 107351 | -90.9404 | 30.0349 | 2.1 | 2.43-2.45 | 2 *Taxodium distichum* cone fragments | 1450±15 | 1.40±0.03 |
| Burton Road I-2 | 109236 | -90.9404 | 30.0349 | 2.1 | 2.19-2.21 | 1 *Taxodium distichum* cone fragment | 945±15 | 0.97±0.01 |
| Burton Road II-1 | 107349 | -90.9454 | 30.0320 | 2.0 | 2.65-2.67 | 3 *Taxodium distichum* cone fragments | 1280±15 | 1.29±0.05 |
| Burton Road II-2 | 107350 | -90.9454 | 30.0320 | 2.0 | 2.67-2.70 | 1 *Nyssa aquatica* stone fragment | 1270±15 | 1.29±0.05 |
| Burton Road III-1 | 109237 | -90.9420 | 30.0338 | 2.0 | 2.56-2.58 | 1 *Nyssa aquatica* stone fragment | 1130±15 | 1.08±0.04 |
| Donner I-1 | 107356 | -90.9570 | 29.6965 | 0.9 | 3.45-3.46 | 1 *Rhynchospora* sp. fruit | 1425±15 | 1.38±0.02 |
| Donner I-2 | 107357 | -90.9570 | 29.6965 | 0.9 | 3.97-3.99 | 2 *Taxodium distichum* cone fragments, 10 charcoal fragments | 2485±15 | 2.66±0.11 |
| Donner II-1 | 137462 | -90.9443 | 29.6967 | 0.5 | 3.91-3.93 | 1 charcoal fragment | 3295±20 | 3.58±0.05 |
| Donner III-1a | 137463 | -90.9403 | 29.6975 | 0.7 | 4.74-4.75 | 1 charcoal fragment | 3925±35 | 4.44±0.13 |
| Donner III-1b | 137464 | -90.9403 | 29.6975 | 0.7 | 4.74-4.75 | 4 charcoal fragments | 3820±25 | 4.28±0.13 |
| *weighted mean age* | | | | | | | *3860±21* | *4.35±0.12* |
| Jeanerette III-1a | 101364 | -91.6360 | 29.9450 | 1.4 | 7.87-7.89 | 1 *Taxodium distichum* cone fragment | 5190±15 | 6.01±0.04 |
| Jeanerette III-1b | 101365 | -91.6360 | 29.9450 | 1.4 | 7.87-7.89 | 1 *Taxodium distichum* cone fragment | 5220±15 | 6.02±0.03 |
| *weighted mean age* | | | | | | | *5205±11* | *6.02±0.03* |
| Loreauville I-1 | 101361 | -91.6813 | 30.0191 | 2.3 | 8.99-9.00 | 2 *Taxodium distichum* cone/twig fragments | 5170±15 | 6.01±0.04 |
| Loreauville II-1a | 101362 | -91.7055 | 30.0215 | 2.5 | 8.68-8.69 | 1 *Nyssa aquatica* stone | 5195±15 | 6.01±0.04 |
| Loreauville II-1b | 101363 | -91.7055 | 30.0215 | 2.5 | 8.68-8.69 | 1 *Taxodium distichum* cone fragment | 5200±15 | 6.01±0.04 |
| *weighted mean age* | | | | | | | *5198±11* | *6.01±0.04* |
| Theriot I-1 | 107358 | -90.7381 | 29.4796 | 0.6 | 4.25-4.26 | 19 charcoal fragments | 965±15 | 0.92±0.07 |
| Theriot II-1 | 107359 | -90.7380 | 29.4789 | 0.5 | 4.84-4.85 | 2 *Scirpus* spp. achenes, 22 charcoal fragments | 900±15 | 0.89±0.07 |

* The age of Amelia I-1 is considered anomalously old and was rejected. † Coordinates were measured with a handheld GPS and rounded to 4 decimals. ‡ All elevations are relative to NAVD 88 and obtained from the National Elevation Dataset 1/3 Arc-second from the USGS (vertical error of ~0.25 m). § Radiocarbon ages were calibrated with the INTCAL13 curve and OxCal 4.1 (Bronk Ramsey, 2009). To facilitate comparison with the OSL ages the mean calibrated age is given relative to AD 2010. The mean age is the midpoint of the calibrated 2σ age range. Since the calibrated 2σ age range is often not symmetrical, the weighted mean age can differ slightly from the mean age.




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
