# Peer review of "Late Holocene evolution of a coupled, mud-dominated delta plain-chenier plain system, Coastal Louisiana, USA"

_Earth Surface Dynamics, 2017_

## Referee Comment (RC1) · A. East (Referee) · 22 May 2017

This paper uses a suite of new OSL and 14C ages to refine the timing of coastal (chenier)-plain development of western Louisiana, USA, downdrift of the Mississippi-Atchafalaya delta, as well as refining chronology of evolution of the delta plain itself in greater detail than any previous study. As the authors state, better understanding the evolution of coupled delta-coastal plain systems has important implications for modeling and projecting the effects of sea-level rise, changes in fluvial sediment output (both affected by climate change) and human activities along the coastal zone and in the watershed. The study is robust and well conceived (and involved a large amount of work), and will be of interest to scientists studying coastal processes, Holocene land-

scape evolution, and fluvio-deltaic stratigraphy and geochronology. This paper should be suitable for publication after minor revisions; most of those I suggest below are intended to expand the discussion of broader context and implications.

The stated hypothesis to be tested, that cyclic Mississippi delta subshifting has influenced evolution of the chenier plain coast, sounds a bit tepid because a number of studies in the 1950s through early 2000s (cited in this paper) already showed pretty convincingly that the evolution of the chenier-plain coast and associated continental shelf are linked to the activity of various sublobes of the Mississippi-Atchafalaya delta. Framing this study's objectives under a more specific hypothesis would set it up so that the results have greater impact (unless the authors intended to test an alternative hypothesis because they didn't believe the findings of those earlier studies, which doesn't seem to be the case).

The introduction presents well the larger scientific and management context of the work, which is substantial in scope and importance. It would be good to remind readers of this big-picture context again at the start of the Discussion section (beyond implications for inferring sea-level history), and revisit the implications further in the Conclusions. As is, the last several sections of the paper are focused so specifically on this immediate region that it may start to lose the broad scientific audience. It is also worth pointing out early in the Introduction (where it is mentioned that few studies address mud-dominated shoreline evolution) that the dynamics of mud-rich coasts are substantially different from those of sand-dominated shorelines, about which much more is known.

The section of the introduction that deals with background information on delta-switching in the Mississippi system needs more complete referencing, e.g., the work of Coleman, additional work by Oscar Huh beyond that cited here, and others. There is a much larger body of literature on this than the text currently reflects.

p. 13, bottom: the "capture" of the mainstem Mississippi by the Red/Atchafalaya River

is better constrained than this. It occurred because a meander bend of the Mississippi (Turnbull's Bend) migrated laterally until it intersected the Atchafalaya during the 15th century, but the full capture has been both inadvertently assisted and now limited in scope by engineering works since the 1830s and especially since the 1950s. This bears mentioning in the text. I also couldn't find mention in the paper of the fact that the Wax Lake Delta formed from an artificially engineered outlet during Atchafalaya River flooding. The authors are certainly aware of these facts, but please state them in the paper for the benefit of readers unfamiliar with this river system.

Late in the Discussion and in the Conclusions, please expand on the potential broader implications for mud-dominated coasts beyond this field area in terms of coastal management, or landscape response to climate change and/or watershed-sediment-supply changes. The Discussion (section 6.1) does go into implications for inferring relative sea level from chenier coasts, which is an advance in understanding the dynamics of mud-rich shorelines and deltas better, and the paper does discuss implications for regional restoration scenarios. . . but can make further contributions by commenting on the additional broader issues just mentioned beyond this geographic region.

Technical corrections:

Section 3.2.1. ends abruptly; it's unclear whether the last sentence was truncated inadvertently.

Figure 14, caption, fix typo: "accumulation rate"

---

## Referee Comment (RC2) · A. D. Switzer (Referee) · 22 May 2017

The new work from Hijma and others uses a collection of new Optically Stimulated Luminescence (OSL) and Radiocarbon (C14) ages to examine and conceptualise the timing of chenier-plain development to the west of the Mississippi delta, USA. The work is comprehensive and scientifically defensible. The work is well illustrated and it will likely be of interest to a wide range of practitioners globally who have interests in fluvio-deltaic and coastal processes, Holocene coastal evolution, sea level change, coastal sediment dynamics and to a lesser extent geochronology.

The work builds nicely on previous studies in the region and considerably improves the chronological framework for the evolution of the delta plain at this important site

thus making it an important contribution. The timeliness of the work is also noteworthy given management plans for the area. The authors are also correct to note that building a detailed insights into the pace and sedimentary characteristics that govern the evolution of coupled delta-coastal plain systems has clear implications for improving management and constraining the effects of sea-level rise, changes in fluvial sediment budgets and human activities in such systems.

This paper is almost suitable for publication after minor revisions however the abstract and introduction both need a bit more bite in terms of why people working elsewhere should care to read and cite this paper. What I would like to see is that the work is placed in a context of some bigger questions. What can we learn from this system that can be applied in other similar systems globally? The work is a little Americo-centric so it would be good to add more international context. What can we glean from this improved understanding for working in similar muddy deltaic systems elsewhere? This should only take a few lines in the introduction and discussion and some additional global references.

Adam Switzer

---

## Referee Comment (RC3) · J. Miselis (Referee) · 16 Jun 2017

In this work, Hijma et al. present a comprehensive dataset documenting the temporal co-evolution of the chenier plain (CP) and the Mississippi delta plain (MDP) in southern Louisiana. Using a combination of detailed sedimentology and stratigraphy and extensive chronological data (OSL and radiocarbon), the work convincingly explains the initiation of CP formation and its evolution resulting from a combination of changes in RSLR, local sediment supply, and the evolution of the MDP. The manuscript skillfully builds on existing literature and offers new insights into the evolution of mud-dominated coastal systems that will surely be of interest globally. With some minor revisions, this

paper will inform scientists interested in the factors that influence the spatial and temporal evolution of fluvio-deltaic systems and coastal resource managers concerned with managing them.

The paper is very successful, but it could be further improved by 1) clarifying the objectives of the paper in the introduction and 2) expanding the discussion to address the global implications of the work. The introduction sets up the reasoning for exploring connections between CP and MDP evolution very well and the corresponding discussion of this relationship is rigorous and thoughtful. However, the discussion begins with a review of the use of cheniers for sea-level reconstruction, which is not clearly established as an objective of the work earlier in the paper. Suggestions for achieving better balance between the introduction and discussion with regard to sea-level reconstruction are included in the technical comments. Finally, the broader implications of the work could use more emphasis. It is completely reasonable to point out the local implications of this study to planned coastal restoration within the MDP, but explicitly identifying other systems that might benefit from the conclusions of this work will broaden the audience.

Technical Comments: Manuscript

Pg. 2, line 17: consider rewording "gain in importance"

Pg. 6, lines 18-20: The vertical error associated with the borehole locations is 0.25m + the variability in elevation within 5-10 m (horizontal accuracy) of the borehole location. The latter component of the vertical error should be easily determined with GIS. Does this influence the interpretation?

Pg. 7, line 3: latest last

Pg. 9, line 2: Is this sample really "anomalously young" or is it just at a higher elevation than the other samples? (and therefore truly younger?) It's difficult to determine the exact elevation from the plots; it would be helpful if the elevations for each sample

(relative to NAVD88) were reported in Table 1 (in addition to surface elevation and depth below surface) to facilitate such comparisons.

Pg. 9, lines 5-8: Is the upper sample in Mura (Creole Ridge) rejected for the same reason above? Is it expected that the base of a chenier and the middle of a chenier would have formed contemporaneously?

Pg. 9, line 26- Pg. 10, line7 and Fig. 6b: It's surprising that there's no discussion of why the JE I-1 sample isn't rejected here given the large 2sigma error (the largest, no?) and that the resulting age does not obey the law of superposition relative to the ages of other JE I and II samples. I realize that the OSL age range of the samples overlaps when the 2sigma error is considered, but I think this is worth mentioning, particularly since similar logic wasn't applied to the rejected samples from the CP. Why are the 2sigma errors are so high for the Jeanerette cross-section relative to all of the other cross-sections?

Pg. 10, line 9: Refer to figures 7 and 8 here.

Pg. 10, lines 25-27 and Fig. 10: Text does not appear to be consistent with figure. The peat bed at -4 to -5m NAVD is clear in Fig. 10, but there aren't any radiocarbon sample locations at the top of this peat bed. There are samples at the top of the peat bed at 0 to -1m NAVD, but these are not what is discussed in the text.

Pg. 12, lines 23-24: Why would erosion in C be a significant source to A, but not to B during the 1.6-1.2ka time period?

Pg. 13, lines 26-27: Add "through" between "halfway" and "the."

Pg. 14: The use of cheniers to construct SLR wasn't really an objective that was laid out in the introduction, but more than 1/3 of the discussion is devoted to it. This point is an important one, but introduce the idea in the introduction. The first paragraph of section 6.1 could be reworked for the introduction.

Pg. 14, lines 9-10: A reference is made to Dougherty et al., 2012, but no explanation

is given as to why this methodology was not employed at the study sites in LA.

Pg. 14, lines 29-31: More explanation of the relationship between Yu et al., 2012 data and the new data is needed here. There is no question that the data fill in a gap in the RSL record, which is exciting. All of the new data, with the exception of 1 point, appear to sit above the Yu et al. data points; only if the values are extrapolated to the extreme end of the error range do they seem to fall in line, undercutting the argument for gradual decrease in RSLR over the last 3ky even if these values are considered maximum limits. Furthermore, given that compaction in the marshy areas is likely to have occurred over the last 2ky, using the modern marsh elevations behind the cheniers is likely underestimating the elevation of the contact between overwash deposits and the marsh behind the chenier. Given these limitations, make your argument for this metric over other metrics stronger. Finally, RSL estimates around 1 ka BP vary by about 1m. What is the explanation for this?

Pg. 15, lines 16-17: Is there a relationship between the area (m2) of headland loss and the increase in downdrift areal gains? If so, presenting this information will help lend support for this argument.

Pg. 15, lines 17-18: Explain why a similar response is not evident in B.

Pg. 17, line 26: slowdown of erosion deceleration of erosion OR decrease in erosion

Technical Comments: Figures

Figs. 3,4,6,8, and 10: Consider adding a legend to each of these figures so readers don't have to flip back and forth between figures.

Figs. 14 and 15: Why not use the same x and y axis orientation for each of these figures?

Technical Comments: Supplementary Material

Fig. S1: The brown color in the cross-section doesn't appear to be in the legend. Also,

"Inner bay" and "Marsh and bay" colors are very difficult to differentiate.

---

## Author Comment (AC1) · 28 Aug 2017

***Response to Reviewer Amy East on*** "Late Holocene evolution of a coupled, mud-dominated delta plain–chenier plain system, coastal Louisiana, USA " **by M.P. Hijma et al.**

**Marc P. Hijma o*n behalf of* Zhixiong Shen, Torbjörn E. Törnqvist, Barbara Mauz**

*We appreciate the review by Amy East. Below our responses to her comments are given in underlined italics.*

The stated hypothesis to be tested, that cyclic Mississippi delta subshifting has influenced evolution of the chenier plain coast, sounds a bit tepid because a number of studies in the 1950s through early 2000s (cited in this paper) already showed pretty convincingly that the evolution of the chenier-plain coast and associated continental shelf are linked to the activity of various sublobes of the Mississippi-Atchafalaya delta. Framing this study's objectives under a more specific hypothesis would set it up so that the results have greater impact (unless the authors intended to test an alternative hypothesis because they didn't believe the findings of those earlier studies, which doesn't seem to be the case).

*Regarding our hypothesis, we disagree with the referee that the hypothesis was convincingly tested. Without the robust chronology that we presented it was also not possible to do so. Our work shows that the hypothesis is only partly valid and local/regional processes played an important role. We write that we want to test the hypothesis more rigorously and we still think that this a good way of describing the main core of our work.*

The introduction presents well the larger scientific and management context of the work, which is substantial in scope and importance. It would be good to remind readers of this big-picture context again at the start of the Discussion section (beyond implications for inferring sea-level history), and revisit the implications further in the Conclusions. As is, the last several sections of the paper are focused so specifically on this immediate region that it may start to lose the broad scientific audience. It is also worth pointing out early in the Introduction (where it is mentioned that few studies address mud-dominated shoreline evolution) that the dynamics of mud-rich coasts are substantially different from those of sand-dominated shorelines, about which much more is known.

*Both at the end of the abstract and the introduction we now highlight the broader implications of this study. In addition, we have added a paragraph in our discussion of the implications for coastal restoration that stresses the importance of using work like we presented here to improve numerical models since these latter will become increasingly important, also globally, in order to save delta from drowning due to sediment mismanagement and relative sea-level rise. Sentences of similar content have been added to the conclusions.*

The section of the introduction that deals with background information on deltaswitching in the Mississippi system needs more complete referencing, e.g., the work of Coleman, additional work by Oscar Huh beyond that cited here, and others. There is a much larger body of literature on this than the text currently reflects.

*Regarding more complete referencing. There is indeed a very large body of literature present on delta-switching in the Mississippi Delta Plain. We chose to refer to the latest review by Blum and Roberts, because it would not be feasible to include all previous work. We propose to add an additional reference to the Coleman et al. review from 1996. These two references should give readers a good starting point if they want to have more background on this topic.*

p. 13, bottom: the "capture" of the mainstem Mississippi by the Red/Atchafalaya River is better constrained than this. It occurred because a meander bend of the Mississippi (Turnbull's Bend) migrated laterally until it intersected the Atchafalaya during the 15th century, but the full capture has been both inadvertently assisted and now limited in scope by engineering works since the 1830s and especially since the 1950s. This bears mentioning in the text. I also couldn't find mention in the paper of the fact that the Wax Lake Delta formed from an artificially engineered outlet during Atchafalaya River flooding. The authors are certainly aware of these facts, but please state them in the paper for the benefit of readers unfamiliar with this river system.

*Capture of the Atchafalaya. We didn't include any details of Turnbull's Bend, because it was not directly necessary information to help us argue that the Atchafalaya River started to have significant sediment output only after 0.3 ka. We added one sentence to introduce Turnbull's Bend and the importance of the removal of a large raft*

Late in the Discussion and in the Conclusions, please expand on the potential broader implications for mud-dominated coasts beyond this field area in terms of coastal management, or landscape response to climate change and/or watershed-sediment-supply changes. The Discussion (section 6.1) does go into implications for inferring relative sea level from chenier coasts, which is an advance in understanding the dynamics of mud-rich shorelines and deltas better, and the paper does discuss implications for regional restoration scenarios: : : but can make further contributions by commenting on the additional broader issues just mentioned beyond this geographic region.
*We included a reference to the Wax Lake Delta.*

Technical corrections:
Section 3.2.1. ends abruptly; it's unclear whether the last sentence was truncated inadvertently.
*We removed the truncated part*

Figure 14, caption, fix typo: "accumulation rate"
*We fixed the typo.*

---

## Author Comment (AC2) · 28 Aug 2017

*Response to Reviewer Adam Switzer on* "Late Holocene evolution of a coupled, mud-dominated delta plain–chenier plain system, coastal Louisiana, USA " *by M.P. Hijma et al.*

**Marc P. Hijma o***n behalf of* **Zhixiong Shen, Torbjörn E. Törnqvist, Barbara Mauz**

*We value the comments of Adam Switzer. His main remark, in line with those from the other reviewers, is to highlight the broader implications of the work much better. We agree that this is necessary. Our response below, in underlined italics, is identical to the response to the comments of the other referees on this issue.*

This paper is almost suitable for publication after minor revisions however the abstract and introduction both need a bit more bite in terms of why people working elsewhere should care to read and cite this paper. What I would like to see is that the work is placed in a context of some bigger questions. What can we learn from this system that can be applied in other similar systems globally? The work is a little Americo-centric so it would be good to add more international context. What can we glean from this improved understanding for working in similar muddy deltaic systems elsewhere? This should only take a few lines in the introduction and discussion and some additional global references.

*Both at the end of the abstract and the introduction we now highlight the broader implications of this study. In addition, we have added a paragraph in our discussion of the implications for coastal restoration that stresses the importance of using work like we presented here to improve numerical models since these latter will become increasingly important, also globally, in order to save delta from drowning due to sediment mismanagement and relative sea-level rise. Sentences of similar content have been added to the conclusions.*

---

## Author Comment (AC3) · 28 Aug 2017

***Response to Reviewer Jennifer Miselis on*** "**Late Holocene evolution of a coupled, mud-dominated delta plain–chenier plain system, coastal Louisiana, USA** " ***by*** **M.P. Hijma et al.**

**Marc P. Hijma** *o**n behalf of* **Zhixiong Shen, Torbjörn E. Törnqvist, Barbara Mauz**

*We thank Jennifer Miselis for her thorough and constructive review. Apart from her technical comments, which we will treat below, she has 2 main recommendations: 1) clarifying the objectives and 2) emphasize the broader implications. Below we give our response to her comments in underlined italics.*

The paper is very successful, but it could be further improved by 1) clarifying the objectives of the paper in the introduction and 2) expanding the discussion to address the global implications of the work. The introduction sets up the reasoning for exploring connections between CP and MDP evolution very well and the corresponding discussion of this relationship is rigorous and thoughtful. However, the discussion begins with a review of the use of cheniers for sea-level reconstruction, which is not clearly established as an objective of the work earlier in the paper. Suggestions for achieving better balance between the introduction and discussion with regard to sea-level reconstruction are included in the technical comments. Finally, the broader implications of the work could use more emphasis. It is completely reasonable to point out the local implications of this study to planned coastal restoration within the MDP, but explicitly identifying other systems that might benefit from the conclusions of this work will broaden the audience.

*Ad 1. We agree that the introduction pays too little attention to the sea-level reconstruction part that constitutes an important aspect of the discussion. We have rewritten the end of the introduction to improve this.*

*Ad 2. Both at the end of the abstract and the introduction we now highlight the broader implications of this study. In addition, we have added a paragraph in our discussion of the implications for coastal restoration that stresses the importance of using work like we presented here to improve numerical models since these latter will become increasingly important, also globally, in order to save delta from drowning due to sediment mismanagement and relative sea-level rise. Sentences of similar content have been added to the conclusions.*

Technical Comments: Manuscript
Pg. 2, line 17: consider rewording "gain in importance"
*We changed this to "become increasingly"*

Pg. 6, lines 18-20: The vertical error associated with the borehole locations is 0.25m + the variability in elevation within 5-10 m (horizontal accuracy) of the borehole location. The latter component of the vertical error should be easily determined with GIS. Does this influence the interpretation?
*Good point. This is especially important in areas, like near the front of the cheniers, where the elevation changes rather rapidly over a short distance. For our geological interpretation it is not important, but it is potentially important for sea-level reconstructions. In our case, we estimated the elevation of the base of the overwash deposit from the cross sections and included an additional error of 0.15 m to account for the spatial variation in the elevation of the base of the overwash, in addition to the 0.25 m error that comes from the DEM. We think that in this way we accounted sufficiently for the vertical uncertainty of the base of the overwash deposits.*

Pg. 7, line 3: latest last

*Changed 'latest' to 'last'*

Pg. 9, line 2: Is this sample really "anomalously young" or is it just at a higher elevation than the other samples? (and therefore truly younger?) It's difficult to determine the exact elevation from the plots; it would be helpful if the elevations for each sample (relative to NAVD88) were reported in Table 1 (in addition to surface elevation and depth below surface) to facilitate such comparisons.

*The elevation of the rejected point is about the same as the other OSL-ages coming from Little Chenier West. We therefore still consider this age anomalously young and did not use it in our calculations. That the age is anomalously young can also be deduced from that the fact that the next seaward chenier, Chenier Perdue, has an age of about 2.6 ka. This means that around 2.46 ka, the age of the rejected sample, Little Chenier no longer formed the shoreline and hence became inactive.*

Pg. 9, lines 5-8: Is the upper sample in Mura (Creole Ridge) rejected for the same reason above? Is it expected that the base of a chenier and the middle of a chenier would have formed contemporaneously?

*The main reason to reject the upper sample is that it shows overdispersion of 20%, a fact that we interpret as the result of post-depositional disturbance and the inclusion of younger grains.*

Pg. 9, line 26- Pg. 10, line7 and Fig. 6b: It's surprising that there's no discussion of why the JE I-1 sample isn't rejected here given the large 2sigma error (the largest, no?) and that the resulting age does not obey the law of superposition relative to the ages of other JE I and II samples. I realize that the OSL age range of the samples overlaps when the 2sigma error is considered, but I think this is worth mentioning, particularly since similar logic wasn't applied to the rejected samples from the CP. Why are the 2sigma errors are so high for the Jeanerette cross-section relative to all of the other cross-sections?

*Pg. 9, line 26- Pg. 10, line7 and Fig. 6b: The error bars around the ages in the Jeanerette section are indeed the largest, but this is mainly due to the fact that they are the oldest OSL-ages in this paper. The relative errors for all OSL-ages are more or less similar and mainly fall between 3-7%, with the samples of JE-II having a relative error 3.5-6%. The JE-I sample indeed has the largest relative error, namely 8.3% and is remarkably young. We have checked our records for this sample and it shows that this sample was taken very close to a boundary between very silty sand and sand in the 30 cm tube that we used for gathering OSL-sample. We initially assumed that our dated sample was taken from the sandy part, but considering the anomalously young age we now think that it is more likely that it comes from the more silty part. This will result in an age of 5.24+/-0.32 ka, which makes more sense. We have changed it throughout the paper.*

Pg. 10, line 9: Refer to figures 7 and 8 here.

*We included the references*

Pg. 10, lines 25-27 and Fig. 10: Text does not appear to be consistent with figure. The peat bed at -4 to -5m NAVD is clear in Fig. 10, but there aren't any radiocarbon sample locations at the top of this peat bed. There are samples at the top of the peat bed at 0 to -1m NAVD, but these are not what is discussed in the text.

*The radiocarbon ages that we refer to are from previous work by Törnqvist et al. (1996) that were obtained by dating the top of a stratigraphically indentical peat bed. We changed the sentence to make this more clear.*

Pg. 12, lines 23-24: Why would erosion in C be a significant source to A, but not to B during the 1.6-1.2ka time period?
*Good point. We think that most of the sediment ended up in A because the headland was sticking out significantly and the plume of eroded sediment could only 'land' a little bit further to the west than segment B. Nonetheless, also in Segment B there is sedimentation due to erosion of the headland, but less significant.*

Pg. 13, lines 26-27: Add "through" between "halfway" and "the."
*We did.*

Pg. 14: The use of cheniers to construct SLR wasn't really an objective that was laid out in the introduction, but more than 1/3 of the discussion is devoted to it. This point is an important one, but introduce the idea in the introduction. The first paragraph of section 6.1 could be reworked for the introduction.
*Very valid point. We changed this by rewriting parts of the introduction.*

Pg. 14, lines 9-10: A reference is made to Dougherty et al., 2012, but no explanation is given as to why this methodology was not employed at the study sites in LA.
*We do write that we consider the base of the cheniers, and hence also the contact between the base and the foreshore deposits, as being problematic to establish a link between chenier formation and contemporary sea level.*

Pg. 14, lines 29-31: More explanation of the relationship between Yu et al., 2012 data and the new data is needed here. There is no question that the data fill in a gap in the RSL record, which is exciting. All of the new data, with the exception of 1 point, appear to sit above the Yu et al. data points; only if the values are extrapolated to the extreme end of the error range do they seem to fall in line, undercutting the argument for gradual decrease in RSLR over the last 3ky even if these values are considered maximum limits. Furthermore, given that compaction in the marshy areas is likely to have occurred over the last 2ky, using the modern marsh elevations behind the cheniers is likely underestimating the elevation of the contact between overwash deposits and the marsh behind the chenier. Given these limitations, make your argument for this metric over other metrics stronger. Finally, RSL estimates around 1 ka BP vary by about 1m. What is the explanation for this?
*We rewrote this section and changed Figure 16 to correct a flaw that was in the submitted paper, namely that the samples plotted in Figure 16 were plotted at the sample elevation instead of the elevation of the base of the overwash deposit. This answers most of the comments by the Referee, since the index points now plot much lower and are more consistent with Yu.*
*It is important to stress that we did not use the modern day elevation of the marsh behind the modern chenier to link our data to past sea level. We only used this elevation to show that the base of the overwash deposit forms above contemporary sea level.*

Pg. 15, lines 16-17: Is there a relationship between the area (m2) of headland loss and the increase in downdrift areal gains? If so, presenting this information will help lend support for this argument.
*Yes there is! The volume of eroded sediment near the headland is between 70-90% of the accumulated volume in segment A. We added a sentence.*

Pg. 15, lines 17-18: Explain why a similar response is not evident in B.
*Same response as earlier: We think that most of the sediment ended up in A because the headland was sticking out significantly and the plume of eroded sediment could only 'land' a little bit further to*

*the west than segment B. Nonetheless, also in Segment B there is sedimentation due to erosion of the headland, but less significant.*

Pg. 17, line 26: slowdown of erosion deceleration of erosion OR decrease in erosion
Technical Comments: Figures
*Changed it.*

Technical Comments: Figures

Figs. 3,4,6,8, and 10: Consider adding a legend to each of these figures so readers don't have to flip back and forth between figures.
*We decided to not do this in order to save space and experience with earlier papers where showing the legend once worked well.*

Figs. 14 and 15: Why not use the same x and y axis orientation for each of these figures?
*Normally, the time is shown on the x-axis. For figure 14 we decided to change this, because we think that in this case this make the figure easier to understand.*

Technical Comments: Supplementary Material

Fig. S1: The brown color in the cross-section doesn't appear to be in the legend. Also "Inner bay" and "Marsh and bay" colors are very difficult to differentiate.
*Good spot. The brown parts signify Marsh and Bay deposits, but the legend is wrong. This also solves the problem to distinguish between Inner Bay and Marsh and Bay.*